# Investigating Generalization Behaviours of Generative Flow Networks

**Lazar Atanackovic**[*]                                              *l.atanackovic@mail.utoronto.ca*
*University of Toronto*
*Vector Institute*

**Emmanuel Bengio**                                              *emmanuel.bengio@valencelabs.com*
*Valence Labs*

**Reviewed on OpenReview:** *https://openreview.net/forum?id=9LOB5N5hUX*

## Abstract

Generative Flow Networks (GFlowNets, GFNs) are a generative framework for learning unnormalized probability mass functions over discrete spaces. Since their inception, GFlowNets have proven to be useful for learning generative models in applications where the majority of the discrete space is unvisited during training. This has inspired some to hypothesize that GFlowNets, when paired with deep neural networks (DNNs), have favorable *generalization* properties. In this work, we empirically verify some of the hypothesized *mechanisms* of generalization of GFlowNets. We accomplish this by introducing a novel graph-based benchmark environment where reward difficulty can be easily varied, $p(x)$ can be computed exactly, and an unseen test set can be constructed to quantify generalization performance. Using this graph-based environment, we are able to systematically test the hypothesized *mechanisms* of generalization of GFlowNets and put forth a set of empirical observations that summarize our findings. In particular, we find (and confirm) that the functions that GFlowNets learn to approximate have an implicit underlying structure which facilitate generalization. Surprisingly —and somewhat contradictory to existing knowledge— we also find that GFlowNets are sensitive to being trained offline and off-policy. However, the reward implicitly learned by GFlowNets is robust to changes in the training distribution.

## 1 Introduction

Generative Flow Networks (GFlowNets, or GFNs) have emerged as a generative modeling framework for learning unnormalized probability mass functions of discrete objects such as graphs, sequences, or sets (Bengio et al., 2021; 2023). They have show particular promise for a wide range of problems and applications where optimization is conducted over very large combinatorial spaces. To name a few, discrete probabilistic modeling (Zhang et al., 2022), molecular discovery (Bengio et al., 2021; Jain et al., 2023), biological sequence design (Jain et al., 2022), and causal discovery (Deleu et al., 2022; 2023; Atanackovic et al., 2023) are areas where the discrete spaces may be intractably large. In these applications, the model will realistically only visit a small fraction of the overall state space. Therefore, understanding how well GFlowNets model and assign probability mass to the unvisited areas of the state space is a critical step. This is fundamentally a question of *generalization* – and is yet to be practically investigated within GFlowNets.

It has been hypothesized that GFlowNets work well because they synergistically leverage the generalization potential of DNNs to assign probability mass in unvisited areas of the state space (Bengio et al., 2021). Indeed, the ability of GFlowNets to learn environment structure plays a critical role in their ability to generalize – a familiar result found in reinforcement learning (RL) (Cobbe et al., 2019; Schrittwieser et al., 2020). Although

---

[*]Part of this work was accomplished during an internship at Valence Labs.
  Our code is available at: `https://github.com/lazaratan/gflownet-generalization`

we have some insight into how GFlowNets work, we know of many failure cases of generalization which exist in RL (Zhang et al., 2018; Packer et al., 2018; Bengio et al., 2020). With the close relationship between GFlowNets and RL (Tiapkin et al., 2024; Deleu et al., 2024; Mohammadpour et al., 2024), we can expect to observe similar problems in GFlowNets when it comes to generalization. Hence, constructing a systematic investigation into generalization within GFlowNets is useful for future algorithmic development, advancement in scientific applications, and our overall understanding.

While the works of Nica et al. (2022) and Shen et al. (2023) probe at the question of generalization within GFlowNets, they only do so superficially. Questions regarding the *mechanisms* of generalization in GFlowNets have yet to be systematically investigated and wholly understood. Thus, unravelling some intuitive notions on *why* and *how* GFlowNets generalize motivates this work. We center our investigation around three primary hypotheses for generalization in GFlowNets:

1. GFlowNets generalize well only **under a narrow set of distributions**, which includes, but is not limited to, sampling from $P_F(s'|s;\theta)$.

2. GFlowNets generalize well because the **objects they are learning have *structure***; $P_F(s'|s)$ and $F(s)$ are not "arbitrary" functions.

3. The difficulty for GFlowNets to generalize is modulated more by the complexity of the reward (**functionally, the generalization error of a supervised DNN**) than the properties of the distribution induced by the reward; **e.g. skewness, temperature**.

To investigate these hypotheses, we devise an empirical investigation under 3 experimental settings. (1) Distilling (regressing to) the *true* flows of the environment. This lets us evaluate generalization on an unseen test set of states and test **Hypotheses** 2 & 3. (2) Measuring memorization gaps when learning the *true* flows while controlling for data and environment structure. Here we can probe the effect of environment structure on generalization to verify **Hypothesis** 2. Finally, (3) training GFlowNets offline and off-policy under different training distributions. This lets us explore the effects of offline-off-policy training and changes in the training distribution on generalization; testing **Hypothesis** 1. These experimental settings help us isolate specific factors of variation which may play a role in *why* GFlowNets generalize.

To conduct our empirical investigation, we propose a set of new graph-based generation tasks to benchmark the performance of GFlowNets for learning unnormalized probability mass functions over discrete spaces. In most existing works, it is common to use the hypergrid and sequence environments to assess performance of GFlowNets in their ability to approximate unnormalized probability mass functions. However, the hypergrid and sequence environments are fundamentally limited for assessing generalization performance of GFlowNets as their structure yields tasks that are easy to learn and approximate. Because of this, we introduce new graph-based benchmark environment where we can incrementally increase the difficulty of the reward while also ensuring that we can easily construct a sufficiently challenging test set of unseen (left out) states to quantify generalization performance. This new environment provides us with a means to more robustly test the generalization capabilities of GFlowNets compared to the commonly used benchmark environments (e.g. the hypergrid and sequence environments).

We describe the details of our experimental setups in §3. In §2, we describe our set of comprehensive benchmark tasks with well defined and tractably computable $p(x;\theta), \forall x \in \mathcal{X}$.[1] Our main contributions are summarized as follows:

- We propose a set of benchmark graph generation tasks of varying difficulty, useful for evaluating GFlowNets' generalization performance.

- We reify and validate some hypothesized characteristics of GFlowNet generalization behaviour over discrete spaces. We accomplish this empirically using benchmark tasks.

---

[1]We note that our proposed empirical study helps us disentangle some of the mechanisms of generalization in GFlowNets, but does not necessarily determine the true underlying causal order of these mechanisms. Nonetheless, we believe this work can pave a road-map for how to approach testing generalization of GFlowNets and view this as a key step towards understanding important properties for this class of algorithms.

- We identify and present a set of observations and empirical findings that form a basis towards disentangling some of the *mechanisms* for generalization of GFlowNets.

## 1.1 Generative Flow Networks

Generative Flow Networks (GFlowNets, GFNs) are a generative modeling framework used to learn to sample from an unnormalized probability distribution. We will refer to this unnormalized function as a positive *reward*, $R(s) > 0$. Introduced in the discrete setting (Bengio et al., 2021), but since then extended to continuous settings (Lahlou et al., 2023), GFlowNets work by learning a sequential, constructive sampling policy. This policy, $P_F$, is used to sample trajectories $\tau = (s_0, ..., s_t)$ where states $s \in \mathcal{S}$ are partially constructed objects, making the state space a pointed directed acyclic graph (DAG) $\mathcal{G} = (\mathcal{S}, \mathcal{A})$ where $(s \to s') \in \mathcal{A} \subset \mathcal{S} \times \mathcal{S}$ is a valid constructive step. There is a unique initial state $s_0$.

While they can be expressed in multiple equivalent ways, we will think of GFlowNets in this work through three main objects: the flow of a state $F(s) > 0$, and the forward and backward policies, $P_F(s'|s)$ and $P_B(s|s')$. In a perfect GFlowNet, these functions are such that for any valid partial trajectory $(s_n, .., s_m)$:

$$F(s_n) \prod_{i=n}^{m-1} P_F(s_{i+1}|s_i) = F(s_m) \prod_{i=n}^{m-1} P_B(s_i|s_{i+1}) \tag{1}$$

where for terminal (leaf) states, $F(s) = R(s)$ by construction. If the above equation is satisfied, then starting at $s_0$ and sampling from $P_F$ guarantees to reach a leaf state $s_t \equiv x$ with probability $p(x) \propto R(x), x \in \mathcal{X}$. We use $\mathcal{X}$ to denote the set of terminal (leaf) states. Note that it may be useful to think of $P_F$ and $P_B$ as representing fractions of flows going forward and backward through the (DAG) network. It may also be useful to think of the flow going through edges: $F(s \to s') = F(s)P_F(s'|s)$.

The so-called *balance condition* in (1) leads to a variety of learning objectives. In this work we primarily use the Sub-trajectory Balance (specifically SubTB(1)) introduced by Madan et al. (2022) and often considered standard, which takes the above conditions, parameterizes the log flow and logits of policies, taking the squared error over all possible subtrajectories of $\tau = (s_0, ..., s_T)$:

$$\mathcal{L}_{\text{SubTB}}(\tau) = \sum_{n < m \leq T} \left( \log \frac{F(s_n) \prod_{i=n}^{m-1} P_F(s_{i+1}|s_i)}{F(s_m) \prod_{i=n}^{m-1} P_B(s_i|s_{i+1})} \right)^2 . \tag{2}$$

We note that trajectory balance (TB) is a special case of the above, where only $n = 0$ and $m = T$ are used. We also note that $F(s)$ is upper bounded (when $P_B$ of all its descendant edges is 1) by the sum of the rewards of all its descendant leaves. For a complete overview of GFlowNets, we refer readers to Bengio et al. (2023). We define terms that we want to make as least ambiguous as possible in §A.

## 1.2 Related Work

**Generalization in deep learning**  Generalization of deep neural networks (DNNs) in supervised learning and deep learning has been extensively studied (Leshno et al., 1993; Pascanu et al., 2013; Zhang et al., 2017; 2021; Kawaguchi et al., 2017). Although it is still not entirely understood, certain mechanisms and intuitive notions for generalization have emerged as the favored schools of thought (Arpit et al., 2017) and inspire this work.

**Generalization in reinforcement learning**  Due to their close relationship with GFlowNets, works studying generalization in Reinforcement Learning also inspire this work (Zhang et al., 2018; Packer et al., 2018; Cobbe et al., 2019; Bengio et al., 2020). For example, generalization behaviours differ when learning the *same* function through regression or through temporal credit assignment (Bengio et al., 2020), which in some form GFlowNets make use of.

**Generalization in GFlowNets**  It is uncontroversial that GFlowNets generalize, to some extent. They learn a distribution that matches the reward in-distribution (Bengio et al., 2021), and on test sets (Malkin et al., 2022), are affected by the choice of parameterization and flow distribution (Shen et al., 2023), and

Table 1: Mean Absolute Error (MAE) when training GFlowNets for different GNN architectures.

| Model | Constant | Counting | Neighbors | Cliques |
|---|---|---|---|---|
| GAT | **0.08** ± **0.00** | **0.14** ± **0.01** | **0.30** ± **0.01** | **0.32** ± **0.03** |
| GCN | 0.48 ± 0.01 | 0.48 ± 0.01 | 2.23 ± 0.19 | 1.53 ± 0.27 |
| GIN | 0.34 ± 0.01 | 0.39 ± 0.01 | 1.48 ± 0.09 | 0.64 ± 0.05 |

Table 2: Jenson-Shannon (JS) divergence when training GFlowNets for different GNN architectures.

| Model | Constant | Counting | Neighbors | Cliques |
|---|---|---|---|---|
| GAT | **0.002** ± **0.000** | **0.002** ± **0.000** | **0.005** ± **0.001** | **0.008** ± **0.001** |
| GCN | 0.031 ± 0.001 | 0.037 ± 0.001 | 0.339 ± 0.017 | 0.240 ± 0.049 |
| GIN | 0.018 ± 0.000 | 0.018 ± 0.000 | 0.247 ± 0.007 | 0.061 ± 0.007 |

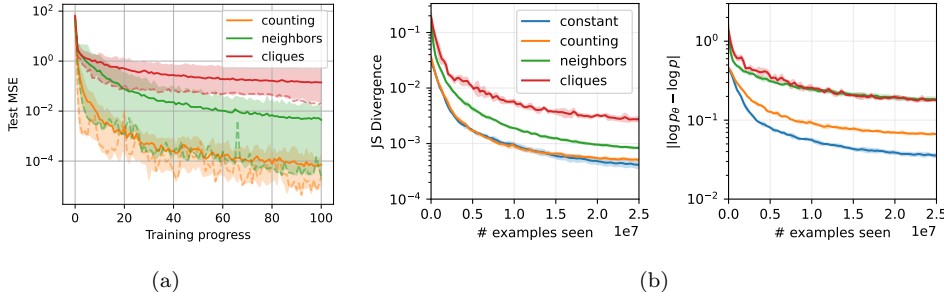

(a)        (b)

Figure 1: (a) Training a GNN on 3 different tasks with models of varying capacity. Most of the variance comes from varying capacity. Dashed lines are for the highest capacity models. (b) Training a GFlowNet (online and on-policy) on 4 different tasks. While ordering is mostly preserved, apparent difficulty depends on the choice of metric.

more generally are able to perform much better than random search in large scale intractable problems. That being said, no study so far has tried to zoom in on whether anything is unique about generalization within GFlowNets.

## 2 A Benchmark Task to Investigate Generalization Behaviours of GFlowNets

As a first step, we present a novel benchmark environment built on a series of graph-based tasks. We use graphs as the foundation of the benchmark tasks as they are a natural choice for compositional discrete objects, and a wide range of combinatorial problems can be expressed as graph generation. We define tasks of varying difficulty on a fixed state space, thus holding the environment constant while the reward difficulty is varied. For completeness, we also conduct experiments on two common benchmark tasks in GFlowNet literature: the **hypergrid** and **sequence** tasks.

### 2.1 A Small Graph Environment

Consider a space of graphs $\mathcal{X}$ and an action space $\mathcal{A} \subset \mathcal{X} \times \mathcal{X}$ of additive graph edits. We then define $P_F(\cdot|s) : A_s \to \mathbb{R}^+; A_s = \{(s, s')|(s, s') \in A\}$. The initial state $s_0$ is the empty graph (no nodes and no edges). The action space has 3 actions: (i) add a new node, (ii) add a new edge, or (iii) stop/terminate. When adding nodes, there is a choice between two "colors" of nodes. This defines the state graph $G = (\mathcal{X}, \mathcal{A})$ for the graph environment; again note that states are themselves graphs.

We setup an environment with all possible graphs of size 7 and less where the nodes can be one of two colors, for a total of 72296 states. This allows us to compute exactly $p(x; \theta)$ for all $x \in \mathcal{X}$ relatively quickly (in the order of 10 seconds on a GPU; see §C.1.2).

**Reward complexity:** We define three different reward functions, which we hope to be of varying difficulty. The hardest function, **cliques**, requires the model to identify subgraphs in the state which are 4-cliques of at least 3 nodes of the same color. **neighbors**, requires the model to verify whether nodes have an even number of neighbors of the opposite color. **counting**, simply requires the model to count the number of nodes of each color in the state. We fully define and show the distribution of $\log R(x)$ of the respective tasks in §C.1. We can observe that the distribution of $\log R(x)$ increases in complexity relative to the hardness of the underlying task.[2]

We verify that our intuitive notion of hardness for these tasks translates to graph neural networks (GNNs). To do so we train standard graph attention networks (Veličković et al., 2017) with varying capacity to regress to the (log) reward functions. We also tried GCNs (Kipf & Welling, 2016) and GINs (Xu et al., 2018), but found GATs to perform significantly better. Because of this, we elected to use GATs for our experiments in this work, as they present the best performance on this graph environment. See Tables 1 and 2 for the ablation over GNN architectures and section D.1 for further details. We use a 90%-10% train-test split, and show the resulting test error in Figure 1(a). We observe that the ordering seems consistent with the difficulty of the task.

**Structural generalization:** The so-called *structural generalization hypothesis* posits that function approximators like DNNs capture regular patterns in the data (Arpit et al., 2017; Zhang et al., 2021), i.e. structure. This enables models to make accurate predictions on similarly structured inputs not present in the training set. We reify this intuition by introducing a structured test set, which we generate by picking states with at least 6 nodes and adding *all their descendants* to the test set, until the test set is of the desired size; see §C.1.3 for details. Generalizing to a "structured" test set this way poses a more challenging problem than simply selecting test samples i.i.d. from the data distribution; this is relevant in domains such as molecular graphs (Tossou et al., 2023), where the choice of split (e.g. based on scaffold) practically matters.

**Metrics for Performance:** To evaluate how well GFlowNets model the distribution $p(x)$, we consider 2 distributional metrics: Jensen-Shannon (JS) divergence and mean absolute error (MAE) between $\log p(x)$ and the learned $\log p(x; \theta)$; see §C.4 for details. Note that when generating graphs, taking into account isomorphic actions is essential to learning the right $p(x; \theta)$ (see Ma et al., 2023, for reference).

**Hypergrid and Sequence Environments:** While the hypergrid and sequence environments have been commonly used to sanity-check GFlowNet implementations and methods, we believe them to be too simplistic to leverage the generalization potential of the DNNs, as will be seen in our experimental results.

## 2.2 Difficulty of Tasks is Preserved When Training GFlowNets Online and On-policy

We verify that training GFlowNets on our proposed graph environment and defined tasks yields the expected task ranking. We train GFlowNets online with SubTB(1) and a uniform $P_B$. To lower bound the modeling complexity, we add a fourth reward, one where $R(x) = 1, \forall x \in \mathcal{X}$.[3] Note that for this specific experiment there are no states hidden from the model, i.e. no test set, since the model is able to sample from its policy and explore the entirety of the state space. Results using test sets are presented in §4.1. Here, we only verify the difficulty of the "usual" GFlowNet setup for these tasks.

In Figure 1(b) we observe that the ordering of tasks is more or less conserved, but depends on the metric we use to measure the discrepancy between $p(x)$ and $p(x; \theta)$. Using a constant reward indeed lower bounds other rewards, but not by much. This suggests that there is, unsurprisingly, inherent difficulty in modeling the dynamics of the environment. We find that there are **two main axes** of difficulty with respect to the task (aside of course from the *scale* of the problem, which here is kept constant): (1) how *difficult* the reward is to model, and (2) how the reward is *distributed*. Hopefully, Figure 1(a) and Figure 1(b) are convincing evidence of (1). We will come back to (2) later in §4.2.

---

[2]For example, the relative quantity of high reward states ("sparseness" of rewards over states) and the general distribution of $\log R(x)$ ("discontinuity" of reward distribution) are factors that can drive learning difficulty.

[3]Note that, hypothetically, this may be harder than other rewards. For a constant reward, the model has to learn to put *equal probability mass everywhere*, which means being able to model the entire state space. In contrast, more complex but "peakier" reward functions may be in some sense easier to "get right" if one cares more about modeling high-reward states, since there are presumably fewer of them.

Readers are hopefully now on board that this benchmark is sufficiently complex, and is useful for empirical validation and hypothesis testing purposes. As mentioned, we also consider 2 non-graph environments, a **hypergrid** environment and **sequence** environment, with reasonably sized states spaces such that we can tractably compute $p(x; \theta)$ exactly. This allows us to investigate GFlowNet generalization behaviours in contexts other than graphs. We use tasks and reward functions used in prior GFlowNet work (Bengio et al., 2021; Malkin et al., 2022; Jain et al., 2023). See §C.2 for details.

# 3 A Method for Disentangling Generalization Mechanisms of GFlowNets

We consider 3 main experimental settings: (1) **distilling (regressing to) flow functions**, (2) **memorization gaps in GFlowNets**, and (3) **offline and off-policy training regimes**. Each experimental setup is built on a series of simplifying assumptions, allowing us control for different moving parts and complexities that are present when training GFlowNets. We now present this experimental protocol, and then report and discuss our findings in §4.

## 3.1 Distilling (Regressing to) Flow Functions

Just as we can compute the distribution $p(x; \theta)$ over $\mathcal{X}$ exactly in this environment, we also compute edge flows $F(s \to s')$ and $P_F(s'|s)$ exactly. We refer to these as the *true* flow and forward policies, in opposition to the approximated $F(s \to s'; \theta)$ and $P_F(s'|s; \theta)$. Note that we parameterize $F(s \to s'; \theta)$ and $P_F(s'|s; \theta)$ as mappings from $\mathcal{S} \to \mathbb{R}^{|n(s)|}$ with $n(s)$ the number of children of $s$.

In this set of experiments, we train DNNs by regressing to the *true* flow[4] $F(s \to s')$ and forward policy $P_F(s'|s)$. This removes the use of the GFlowNet training objective, and instead of the training distribution coming form trajectories sampled from $P_F(s'|s; \theta)$ we sample $s$ from the training set, drastically simplifying the training procedure. As such, we basically control for any non-ideal factors within the GFlowNet training setup, such as shortcomings due to temporal credit assignment (Malkin et al., 2022).

To regress to $\log F(s \to s')$ or $\log P_F(s'|s)$, both vectors, we regress to each value independently and minimize the mean squared error. Note that the true $P_F(s'|s) = \text{softmax}(\log F(s \to s'))$. Regressing to $\log F(s \to s')$ is thus almost like regressing to $\log P_F(s'|s)$, but the model has to get the absolute magnitudes of each logit right, not just the relative ones (the softmax in $P_F$ normalizes). Because of this, we expect that learning $P_F(s'|s)$ is easier, although learning flow magnitudes could help with generalization. We set $P_B$ to be the uniform policy to get unique $F$ and $P_F$.

Using this setup we are able to assess how fundamentally difficult these flow functions are to learn. Furthermore, we can use our test set of unseen states. This allows us to assess generalization performance for learning $p(x)$ (which we are able to tractably compute) by directly computing distributional errors. Because of the use of a test set, this is in some sense more challenging than when training GFlowNets directly, i.e. online and on-policy, since standard training of GFlowNets allows the model to potentially explore the entire space and "see" the entire dataset[5]

Overall, in this setting we can probe the question of "*do*" GFlowNets generalize when learning $F$ or $P_F$ and assess **Hypotheses** 2 & 3. In the following sub-section, we describe a second approach that can help us investigate some mechanistic intuitions of *why* flow functions might induce generalization in GFlowNets, probing the question of "*why do*" GFlowNets generalize.

## 3.2 Memorization Gaps in GFlowNets

We would like to probe our hypothesis on the contribution to generalization of learning flows in GFlowNets. We take the perspective of Zhang et al. (2017; 2021) and consider generalization as the act of *not memorizing*. In other words, we can assess whether a model is generalizing or not by measuring the gap (in *training*

---

[4]In this sense, we are "distilling" the *true* flows into flow functions parameterized by a DNN.

[5]Readers may notice that because we've computed the regression targets $F$ and $P_F$ exactly using the entire state space, some information of the test set is leaking into the targets. Since the purpose of this experiment is to assess how hard these functions fundamentally are to learn, we knowingly allow the model to cheat.

performance) between training it on structured data and training on random unstructured data. We use this notion of *memorization gap* to examine how learning flows with or without structure impacts generalization.

We devise an experiment inspired from Zhang et al. (2017; 2021), where we train supervised models by regressing to $R(s)$ and $P_F(s'|s)$ (using the framework described in the previous section), but with various degrees of "de-"structuring. Consider simply learning to predict $R(s)$ with $R(s; \theta)$. If instead of regressing from $s$ to $R(s)$ we shuffle the reward labels of each state $s$, we end up with new $(s, \tilde{R}(s))$ pairings. This induces independence between the data pairings, i.e. $s \perp\!\!\!\perp \tilde{R}(s)$. Intuitively, training a DNN with $s \perp\!\!\!\perp \tilde{R}(s)$ will force it to memorize, having removed structure in the mapping from $s$ to $R$.

Table 3: High level summary for the memorization gap experiments. Each row lists an individual experiment showing the corresponding data pair coupling and structure of learning problem. These experiments consider models trained with distilled/regressed learning (not via online training using SubTB(1)). $\tilde{P}_F(s'|s)$ denotes the policy logits derived from the shuffled rewards.

| Learning (Regressing to) | Data Coupling | Reward (data) Structure | Flow (environment) Structure |
|---|---|---|---|
| $R$ | $s \not\perp\!\!\!\perp R(s)$ | ✓ | ✗ |
| $R$ | $s \perp\!\!\!\perp \tilde{R}(s)$ | ✗ | ✗ |
| $P_F$ | $(s, s') \not\perp\!\!\!\perp P_F(s'|s)$ | ✓ | ✓ |
| $P_F$ | $(s, s') \not\perp\!\!\!\perp \tilde{P}_F(s'|s)$ | ✗ | ✓ |
| $P_F$ | $(s, s') \perp\!\!\!\perp P_F^{\text{random}}$ | ✗ | ✗ |

Now consider regressing to $P_F(s'|s)$; there are two ways to destructure this function. Recall that $F(s)$ (and so implicitly $P_F(s'|s)$) is a function of the *reward* of all its descendants as well as of the *transition structure* of the state space (the ways to get to those descendants). Let $\mathcal{D}(s)$ be the set of descendants of $s$. By shuffling $R$ into $\tilde{R}$, we thus remove the dependence between $s$ and $\tilde{R}(s_d) \forall s_d \in \mathcal{D}(s)$, but keep the dependence between $s$ and *how to get to* $\mathcal{D}(s)$. The second way to de-structure $P_F$ is to simply regress to random logits, which we will denote $P_F^{\text{random}}$.

To recap, we regress to: $R(s)$ using paired data, $R(s)$ using shuffled data, $P_F(s'|s)$ using paired data, $P_F(s'|s)$ using shuffled data, and $P_F(s'|s)$ using random logits $P_F^{\text{random}}$ from a magnitude-preserving range. This is summarized in Table 3.

Through this setup, we can assess the *memorization gap* that occurs when training models. Destructuring the reward but maintaining flow structure in the learning problem allows us to assess the contribution that learning *true* flows has on generalization (i.e. *not memorization*) when training GFlowNets. Following from **Hypothesis** 2, if learning structured flows reduces the degree of memorization, perhaps flow prediction inherently acts as a mechanism for generalization in GFlowNets.

### 3.3 Offline and Off-policy Training Regimes

Lastly, we investigate the effects of deviating from the self-induced training distributions of GFlowNets on generalization. To do this, we consider the setting of training GFlowNets *offline* and off-policy given a known dataset of final states $\mathcal{X}$. We sample $x \sim \mathbb{P}_{\mathcal{X}}$, where $\mathbb{P}_{\mathcal{X}}$ is some training distribution over $\mathcal{X}$. We consider different distributions for $\mathbb{P}_{\mathcal{X}}$ that resemble dif-

Table 4: Details for different training distributions $\mathbb{P}_{\mathcal{X}}$ for offline and off-policy experiments.

| | $\mathbb{P}_{\mathcal{X}}$ | Resemblance to Sampling $x$ |
|---|---|---|
| **Uniform** | $\propto \mathcal{U}(x)$ | i.i.d. |
| **Log-rewards** | $\propto R(x)$ | from an ideal policy |
| **Proxy for on-policy** | $\propto p(x; \theta)$ | $\propto$ to $P_F(s'|s; \theta)$ |
| **Absolute error** | $\propto |p(x; \theta) - p(x)|$ | $\propto$ absolute loss |
| **Squared log-error** | $\propto (\log p(x; \theta) - \log p(x))^2$ | $\propto$ squared log-loss (e.g. SubTB(1)) |

ferent practical approaches and techniques that are used for training GFlowNets (see Table 4). This setting controls for the effect of sampling from $P_F(s'|s; \theta)$ in training GFlowNets.

Given a terminal $x$, we sample a trajectory $\tau = (s_0, \ldots, s_f)$ backwards using a uniform $P_B$[6]. Hence, we term this as *offline* training, since we are avoiding sampling $x$ using $P_F(s'|s; \theta)$. We use a standard GFlowNet objective, SubTB(1), to train $P_F(s'|s; \theta)$ using those $\tau$. Note that this setup can remove the data non-stationarity that normally exists when using on-policy $P(s'|s; \theta)$ samples.

We also consider the online off-policy setting, to assess to what degree deviating from $P_F(s'|s; \theta)$ during training will affect learning $p(x)$. Understanding this may be useful to develop novel GFlowNet algorithms,

---

[6]Investigating the use of different $P_B$s is another interesting direction that may give insight into the effects of the backward policy on the training dynamics of GFlowNets (see Shen et al., 2023). We leave this for future work.

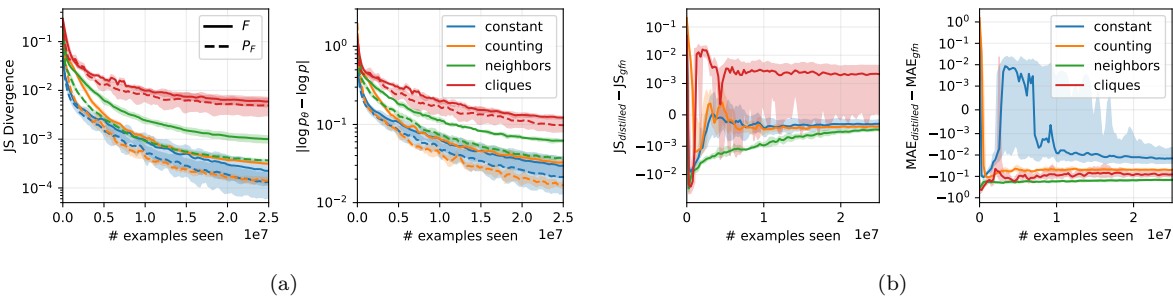

Figure 2: (a) Training a model to distill edge flows and policies. (i) doing so recovers the intended distribution, (ii) modeling $P_F$ appears easier than modeling $F$. (b) JS-divergence and MAE gaps between SubTB(1) trained model and $P_F$ distilled model. Distilling to $P_F$ appears to yield lower distributional error, except for JS-divergence on the cliques task.

considering that in practice only few settings that deviate from on-policy training work (Rector-Brooks et al., 2023). To achieve this, we consider a policy interpolation experiment where we sample trajectories from $P_\alpha(s'|s) = (1-\alpha)P_F(s'|s;\theta) + \alpha P_\mathcal{U}(s'|s)$, where $0 \leq \alpha \leq 1$ and $P_\mathcal{U}(s'|s)$ denotes a $P_F$ that goes to every terminal state with equal probability, i.e. such that $p(x) \propto 1$. We do this because, as we will see, any degree of deviation from $P_F(s'|s;\theta)$ during *offline* training appears to have some negative effect. We observe this even for values of $\alpha$ close to 0. We discuss this in further detail in section 4.4.

This experimental setup helps us investigate the isolated effect of different data distributions $\mathbb{P}_\mathcal{X}$ on training dynamics and generalization by changing the data distribution (and non-stationarity) typically induced by $P(s'|s;\theta)$. We consider both settings with and without a test set. This allows us to assess **Hypothesis** 1 by observing the effects of various deviations from the normal self-induced training distribution coming from $P_F(s'|s;\theta)$.

## 4 Experiments

In this section, we report and discuss our observations and findings of the experimental methodology defined in §3. We conduct all experiments reported in this section over 3 random seeds. For training online and offline GFlowNets, we use SubTB(1) (see §D.2) and a uniform $P_B$.

### 4.1 Distilling Flow Functions

We run the setup explained in §3.1. To evaluate the models, we look at the distributional errors on $p(x;\theta)$. We use a 90%-10% train-test split. We report results in Figure 2(a) and Figure 2(b).

First, we can see that learning $P_F$ and learning $F$ yield similar difficulty in the sense that the distributional errors are systematically close. Second, we see that the model is generalizing, getting sometimes even better scores than a model trained online on the entire states space (i.e. with no test set) as seen in Figure 2(b). While this result should be obvious, it confirms that the difficulty of training a GFlowNet comes from both learning to model the flow functions themselves *and* performing temporal credit assignment.

> **Observation 4.1.** *Flow functions and flow policies are learnable and standard neural networks generalize when predicting them.*

We confirm these observation in sequence and grid environments (see §E.1). Further results in §D.3 assess how well GFlowNets (and TB Malkin et al. (2022) and FM (Bengio et al., 2021)) generalize *in rank* in the graph tasks.

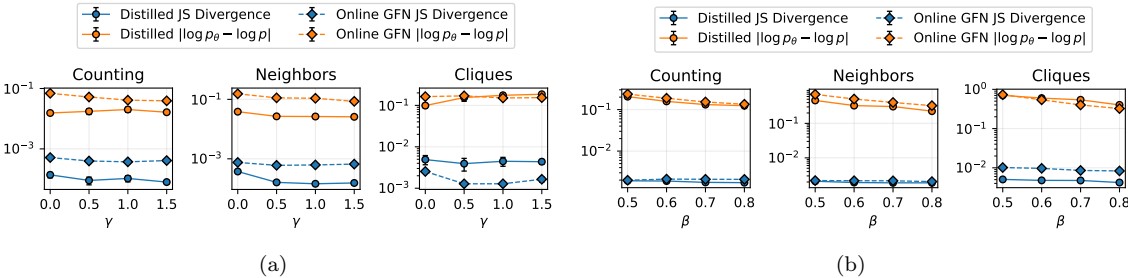

(a)                                             (b)

Figure 3: (a) Training models distilled to $P_F$ and models trained online/on-policy for a range of $\gamma$; (b) similarly for a range of $\beta$. Transforming the distribution of the reward does not significantly affect the generalization difficulty in approximating $p(x)$.

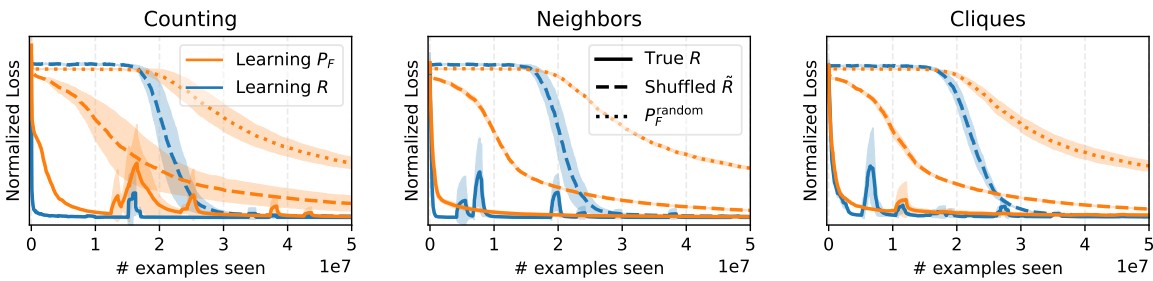

Figure 4: Memorization gap training curves for counting, neighbors, and cliques tasks. Maintaining *flow* structure in the learning problem (learning $P_F$ under shuffled $\tilde{R}$) induces a smaller *memorization gap* relative to the fully de-structured setting. See Table 3 for reference of experimental setup.

## 4.2 Reward complexity and transformations

Consider a monotonic transform $\mathcal{H}_\gamma : \mathbb{R} \to \mathbb{R}$ that maps $\log R'(s) = \mathcal{H}_\gamma(\log R(s)), \forall s \in \mathcal{S}$ such that $\log R'(s)$ is skewed as a function of $\gamma$ (see §C.3 for further details). We use the setup in §3.1 and also train online GFlowNets in the setting of skewed reward distributions transformed by $\mathcal{H}_\gamma$ as well as tempered distributions via a $\beta$ parameter, $\log R'(s) = \beta \log R(s)$. Results are in Figure 3(a); as we change $\gamma$ and $\beta$, both distilled and online models are not significantly affected.

> **Observation 4.2.** *Monotonic transformations, such as skew and reward tempering, do not have detrimental impact on generalization when learning $p(x)$ with GFlowNets, for the range of hyperparameters considered.*

This implies that applying minor reward transformations, which is a standard technique in GFlowNet training, is not detrimental to *generalization* about $p(x)$ for a given task (of course, e.g. sparsifying a reward makes exploration harder), supporting **Hypothesis** 3. We repeat this experiment on the sequence environment and find consistent results (see §E.2).

## 4.3 Memorization Gaps in GFlowNets

We run the setup explained in §3.2 and Table 3; results are shown in Figure 4. Additional results with $R(s) \sim \mathcal{N}(\mu, \sigma)$ and online trained GFlowNets are in §D.5. A typical observation in the *memorization gap* setup is for the training loss to initially plateau, until some phase change occurs (one can imagine parameters have self-arranged to create separate linear regions around each input point) and the training loss goes to 0. We se similar trends here, regressing to $R$ and $P_F$ induce curves with no plateau, while regressing to $\tilde{R}$ and $P_F^{\text{random}}$ initially have very flat plateaus until a phase change occurs. This shows a very clear memorization gap, which was expected.

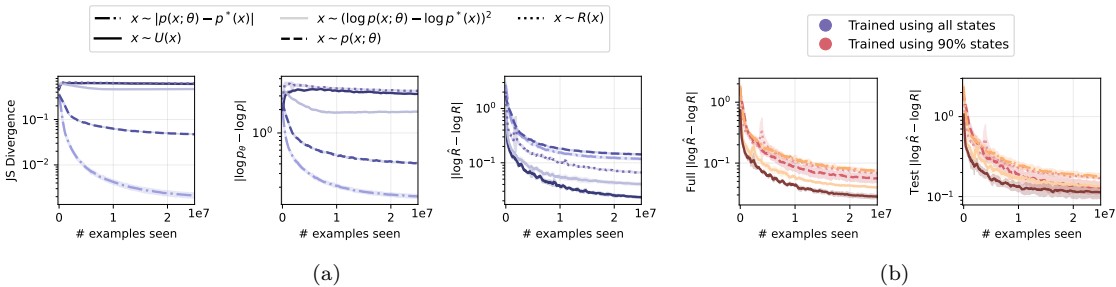

Figure 5: Evaluation curves for offline and off-policy trained GFlowNets on the neighbors task for different choices of $\mathbb{P}_{\mathcal{X}}$. (a) When training using the full dataset (no test set). (b) When training using a subset of the full dataset (90%-10% train-test split). Complete experiments for all graph generation tasks and evaluation metrics are shown in §D.6.

Most interesting is where $P_F$ is *partially* de-structured, and we regress to the *true* $P_F$ of a shuffled reward $\tilde{R}$ and retain the environment structure. While there is no plateau, $P_F$ becomes harder to fit, and there remains a phase change. These results support **Hypothesis** 2; $P_F$ captures structure of both the reward function and the environment, since removing either induces *memorization gaps*.

> **Observation 4.3.** *The existence of clear memorization gaps, when removing reward structure but maintaining environment structure, upholds that environment structure facilitates generalization when learning flow functions and flow policies.*

This finding implies that state flows may play a mechanistic role in *why* GFlowNets generalize to unseen states. We also show similar results for sequences and grids (see §E.3).

## 4.4 Generalization in Offline and Off-policy Training of GFlowNets

We run the offline training setup described in §3.3 and Table 4; results on the neighbors task are shown in Figures 5(a) and 5(b). Results for all tasks are in §D.6. First, we observe that performance is dependent on environment/task difficulty, the choice of $\mathbb{P}_{\mathcal{X}}$, and on the evaluation metric. For example, the best choice for $\mathbb{P}_{\mathcal{X}}$ on the counting task is typically $x \sim \mathcal{U}$ or $x \sim \log R(x)$, while this is not always the case for neighbors and cliques (see §D.6 Figure 13).

Second, we observe that when training $P_F(s'|s;\theta)$ offline and off-policy in the full data (no test set) regime, convergence rate heavily depends on the choice of $\mathbb{P}_{\mathcal{X}}$ (see Figure 5(a)), sometimes to a catastrophic degree. We see this for our usual distributional metrics for approximating $p(x)$[7]. However, when considering the MAE between $\log R(s)$ the approximated $\log \hat{R}(s)$, we do not observe this lack of convergence (GFlowNets implicitly learn to predict the reward: $\log \hat{R}(s) = \log F(s;\theta) + \log P_F(s_f|s;\theta)$). Although training $P_F(s'|s;\theta)$ offline or off-policy may be tricky, learning $F(s;\theta)$ jointly with $P_F(s'|s;\theta)$ might be beneficial: we observing a reasonable estimate for $\log \hat{R}(s)$ even when $P_F(s'|s;\theta)$ fails to adequately model $p(x)$—the model is learning something.

We then run the off-policy interpolations (as described in §3.3), with $P_\alpha = (1 - \alpha)P_F + \alpha P_{\mathcal{U}}$ and plot the results in Figure 6. We observe a trade-off between on-policy and off-policy training when estimating $p(x)$ versus $R$. We find that for harder tasks (neighbors, cliques), making the data more off-policy ($\alpha = 1$) makes $p(x;\theta)$ become worse and $\hat{R}$ better. When sampling on-policy (from $P_F(s'|s;\theta)$, $\alpha = 0$), approximating $p(x)$ is reasonable, but approximating $R$ is best when sampling uniformly off-policy (from $P_{\mathcal{U}}(s'|s)$, $\alpha = 1$).

> **Observation 4.4.** *Deviating from the on-policy training distribution induced by $P_F(s'|s;\theta)$ can negatively affect the ability of GFlowNets' to learn $p(x)$.*

---

[7]It is possible that the drawbacks of this training regime overwhelm our ability to make meaningful conclusions on the choice of $\mathbb{P}_{\mathcal{X}}$ for offline training on the neighbors and cliques tasks.

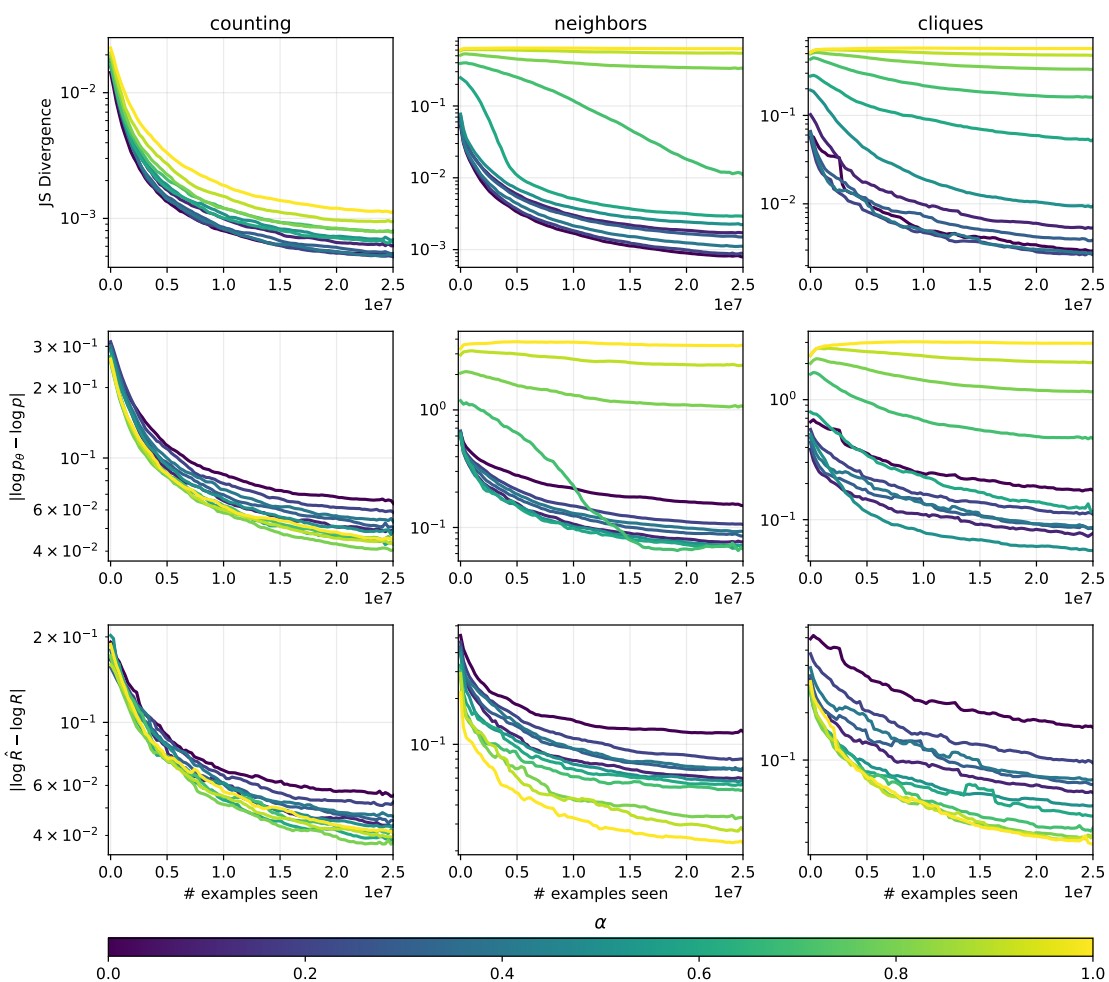

Figure 6: Results for policy mixing interpolation experiments. Here, we mix $P_\alpha(s'|s) = (1-\alpha)P_F(s'|s;\theta) + \alpha P_\mathcal{U}(s'|s)$, where $0 \leq \alpha \leq 1$. For $\alpha = 0$, the model is fully on-policy, sampling trajectories from $P_F(s'|s;\theta)$. For $\alpha = 1$, the model is sampling fully off-policy using $P_\mathcal{U}(s'|s)$. This approximates the behaviour of training offline and off-policy when using $x \sim \mathcal{U}(x)$ (i.e. uniform sampling of $x$). Here we can observe the effects of deviating from on-policy samples during training. We see that at times, specifically for the more difficult tasks (neighbors and cliques), even small degrees of deviation from $P_F(s'|s;\theta)$ can lead to worsened performance when approximating $p(x)$ (seen by sensitivity in distributional JS divergence and MAE metrics). In contrast, approximating $R$ appears to improve as the model samples more uniformly over $\mathcal{X}$ (if not entirely uniform, i.e. when $\alpha = 1$). Interestingly, the counting task appears generally invariant to minor changes in $\alpha$.

This implies that training GFlowNets offline or off-policy may come with challenges. However, the model could be learning some informative structure in $F(s)$ that helps it generalize (seen by a reasonable $\log \hat{R}(s)$). We test this hypothesis in our second offline and off-policy training setup, and consider performance on a test set (Figure 5(b)). We observe that offline and off-policy trained GFlowNets reasonably approximate $R$ on unseen states even when failing to adequately model $p(x)$.

> **Observation 4.5.** *Learning $F(s)$ can facilitate models learning to implicitly predict the reward on unseen states, even when $P_F$ does not adequately approximate $p(x)$ on these states.*

These results support **Hypothesis** 1; training GFlowNets online and on-policy is ideal, where $P_F(s'|s;\theta)$ converges to sampling in proportion to $R$. We repeat these experiments for the sequence and hypergrid

environments, where we observe that $p(x)$ is adequately modelled in the offline and off-policy cases (see §E.4). This may be due to the lesser complexity of these environments, which we establish by comparing the difficulty of our tasks to that of fitting a constant reward (see §E.1).

## 5    Conclusion

In this work, we conducted an empirical investigation into the generalization behaviours of GFlowNets. We introduced a set of graph generation tasks of varying difficulty to benchmark and measure GFlowNet generalization performance. Our findings support existing hypothesized *mechanisms* of generalization of GFlowNets as well as add to our understanding of *why* GFlowNets generalize. We found that GFlowNets inherently learn to approximate functions which contain structure favourable for generalization. In addition, we found that the reward implicitly learned by GFlowNets is robust to changes in the training distribution, but that GFlowNets are sensitive to being trained off-policy.

**Discussion:**   Our objective in this work was to take a scientific approach of starting from existing set of hypotheses and testing them through a more comprehensive set of benchmark tasks. For example, many existing works claim that GFlowNets can effectively operate in the off-policy regime. However, through our investigation, we show that GFlowNets are not as robust as the tone of those claims would suggest. We demonstrated this by introducing a novel graph-based benchmark environment where task difficulty can be easily varied while also varying the off-policy sampling distribution. We repeated our empirical investigation using standard environments which are used in GFlowNet literature (i.e. the hypergrid and sequence environments). Interestingly at times, and particularly for the off-policy results, there is a discrepancy of the observed behavior of GFlowNet generalization performance between the hypergrid and sequence environments with performance on our graph-based environment. This result suggests the importance of taking into consideration the choice of benchmark environment when developing novel GFlowNet methods.

Our intention was to systematically test existing hypotheses regarding GFlowNets and their ability to generalize to unseen area of the state space. In testing these hypotheses, we confirmed existing knowledge while also finding contradictory outcomes from what has been previously understood. For example, while Observations 4.1-4.3 confirmed our existing received knowledge regarding GFlowNets and their generalization capabilities, Observations 4.4 and 4.5 were unexpected results. We believe these are interesting findings that warrant further investigation and could for example lead to improvements in off-policy training and exploration algorithms for GFlowNets.

**Limitations:**   We have shown that GFlowNets tend to generalize when learning unnormalized probability mass functions for approximating $p(x)$ over discrete spaces. In particular, we investigated the generalization behaviours of GFlowNets in the context of distributional errors for $p(x)$. Because of the combinatorial nature of computing $p(x)$ exactly, and likewise computing $p(x; \theta)$ from $P(s'|s; \theta)$, we are limited in the fact that we need to work within reasonably sized discrete spaces (as those we have proposed) to study GFlowNet generalization for approximating $p(x)$. Secondly, we don't explicitly test generalization in online and on-policy trained GFlowNets. This in part a consequence of requiring an environment and state space large enough such that a sufficient quantity of unseen states can be produced, while also being able to purposely hide the visited states from parameters updates of the GFlowNet. Lastly, although our proposed problems and environments have structure that induces difficult and interesting generalization tasks, they still may remain far from the structure of the real world.

**Future Directions:**   We hope to have formed a sound set of findings and observations that may lead to future research in understanding and formalizing generalization in GFlowNets. For instance, our set of findings could be used as starting points for formalizing some of our notions and intuitions for GFlowNet generalization (and the corresponding mechanisms) into mathematical theory. Another direction that can stem from our work is to further empirically investigate GFlowNet generalization in the online and on-policy setting. Since we don't explicitly look at GFlowNet generalization in the online and on-policy training regime (i.e. since we don't hide any states from online trained GFlowNets); it would be interesting in future work to explicitly investigate this. However, it is important to note that this is not necessarily a trivial task (as described in limitations).

## Acknowledgments

The bulk of this research was done at Valence Labs as part of an internship, using the computational resources there. The authors are grateful to Yoshua Bengio, Berton Earnshaw, Dinghuai Zhang, Johnny Xi, Jason Hartford, Philip Fradkin, Cristian Gabellini, Julien Roy, and the Valence Labs team for fruitful discussions and feedback. In addition, we acknowledge funding from the Natural Sciences and Engineering Research Council of Canada and the Vector Institute.

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

# A   Vocabulary

- **The $p(x; \theta)$ distribution**, when using GFNs we never explicitly model $p(x; \theta)$; instead this measure is induced by the parameterization ($F(s \to s'; \theta)$ or $P_F(s'|s; \theta)$) of a sequential constructive policy. We use $p(x; \theta)$ as the shorthand for *the distribution over $X$ induced by the chosen parameterization with parameters $\theta$*. See also §C.1.2.

- **Online vs offline**, a model is trained online when it is trained from data generated during the training process – typically according to its own parameters, i.e. for some model $p(x; \theta)$, we may use $X \sim p(x; \theta)$ to train $p(x; \theta)$. Conversely a model trained offline is trained from a (usually fixed) data set, $X \sim \mathcal{D}$. It is possible to form a mixture of online and offline data. GFNs are compatible with this paradigm (unlike RL methods such as policy gradient methods).

- **On-policy vs off-policy**, a model is trained on-policy if it is trained from data generated according to the unperturbed distribution $p(x; \theta)$, whereas it is trained off-policy if it comes from any other distribution. For example, taking some actions at random would be considered off-policy (although a mild version), but so would taking samples from an entirely different distribution $p(x; \theta')$ or from a dataset.

- **Self-induced distribution**, when training a model we refer to the distribution $p(x; \theta)$ as "self-induced". This is mainly to emphasize that this distribution changes as $\theta$ changes, which in online on-policy contexts is due to the model generating its *own* training samples (thus the "self"-induced).

- **True flows**, we occasionally refer to *true flows*. What we mean by that is the exact calculation of $F(s)$ or $F(s \to s')$ as a function of the reward $R(s)$ and of $P_B$. Note that for a fixed $P_B$ there is a unique solution to $F$ (Bengio et al., 2021). When the reward is corrupted, we talk about the true flow of the corrupted reward $\tilde{R}$, i.e. $F(s)$ is the exact calculation of the flow through $s$ but for terminal (sink) flows set to $\tilde{R}$.

# B   Limitations and Future Work

**Impact Statement**   The outcomes of the research conducted in this work have practical implications for generative modeling in applications that include drug discovery, material design, and general combinatorial optimization in commercial settings. Because of this, we recognize the importance of considering safety and alignment in these closely related domains. We believe that advancements in GFlowNet research could result in models with improved generalization, in turn leading to models that are easier to align. Another important and useful application of GFlowNets is in the causal and reasoning domains. We believe that advancements in these areas may lead to more interpretable, easier to understand, and safer models. Lastly, we highlight that the applications of our findings for the advancements of GFlowNets may rely on building upon the hypotheses put forward in this work.

# C   Experimental Details

## C.1   Details for constructing graph benchmark task

### C.1.1   Reward functions

Here are the exact log-reward functions we use, as implemented in Python using the `networkx` (`nx`) and `numpy` (`np`) libraries.

**cliques** counts the number of 4-cliques in a graph such that at least 3 of the 4 nodes of the cliques share the same color. We then subtract to that number the total number of cliques in the graph, but add the number of nodes. This is so that the maximal log-reward is 0. We clip log-rewards below -10.

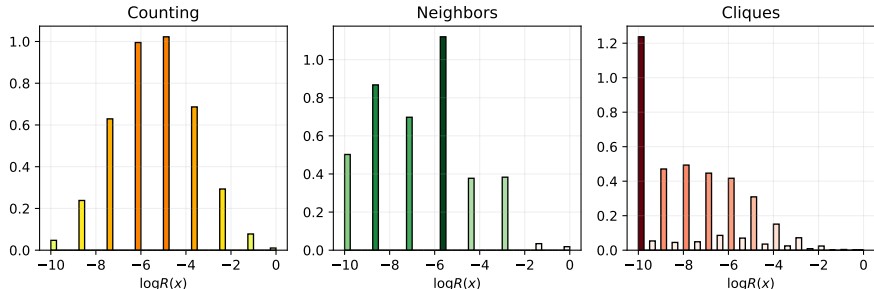

Figure 7: Distribution of normalized count of log-rewards over the state space for the given tasks: (left) **counting**, (middle) **neighbors**, and (right) **cliques**.

```
def cliques(g, n=4):
    cliques = list(nx.algorithms.clique.find_cliques(g))
    # The number of cliques each node belongs to
    num_cliques = np.bincount(sum(cliques, []))
    colors = {i: g.nodes[i]["v"] for i in g.nodes}

    def color_match(c):
        return np.bincount([colors[i] for i in c]).max() >= n - 1

    cliques_match = [float(len(i) == n) * (1 if color_match(i) else 0.5) for i in cliques]
    return np.maximum(np.sum(cliques_match) - np.sum(num_cliques) + len(g) - 1, -10)
```

**neighbors** looks at all the neighbors of every node, counting the number of nodes with an even number of neighbors of the opposite color. This total is modified as a function of the number of nodes in order to produce a nice log-reward distribution between 0 and -10.

```
def neighbors(g):
    total_correct = 0
    for n in g:
        num_diff_colr = 0
        c = g.nodes[n]["v"]
        for i in g.neighbors(n):
            num_diff_colr += int(g.nodes[i]["v"] != c)
        total_correct += int(num_diff_colr % 2 == 0) - (1 if num_diff_colr == 0 else 0)
    return np.float32((total_correct - len(g.nodes) if len(g.nodes) > 3 else -5) * 10 / 7)
```

**counting** simply counts the number of red and green nodes, red nodes being "worth" more, and again modifies this count in order to produce a nice 0 to -10 log-reward distribution.

```
def counting(g):
    ncols = np.bincount([g.nodes[i]["v"] for i in g], minlength=2)
    return np.float32(-abs(ncols[0] + ncols[1] / 2 - 3) / 4 * 10)
```

### C.1.2 Computing $p(x;\theta)$, $F(s)$, $F(s \to s')$, and $P_F$

We compute the probability of a model with parameters $\theta$ sampling $x$, $p(x;\theta)$ via what is essentially Dynamic Programming. We visit states $s$ in **topological order** starting from $s_0$. For every child of $s$, $s'$, (valid transition $s \to s'$) we add $p_v(s)P_F(s'|s;\theta)$ to the probability of visiting $p_v(s')$, eventually accumulating visitation probability from all the parents of $s'$. This is possible because we are visiting states in topological order. Note that we start with $p_v(s_0) = 1$ and $p_v$ set to 0 for every other state. $p(s;\theta)$ is computed as $p_v(s)P_F(\text{stop}|s)$.

The above is fairly easy to batch, requires one forward pass per state. Computing $P_F$ is the most expensive operation and can be batched, as long as values are accumulated by respecting a topological order afterwards. We also actually store values on a log scale, using `logaddexp` operations for numerical stability.

To compute *true* flow functions, we do the reverse, visiting the DAG in reverse topological order. For every state $s$, we add $F(s)P_B(s'|s)$ units of flow to every parent $s'$ of $s$, starting at leaves (due to the reverse topological order) where $F(s) = R(s)$. We similarly set the value of the edge flow $F(s \to s') = F(s)P_B(s'|s)$. $P_F$ is simply the softmax of edge flows. Note that we use a uniform $P_B$, as explained in the main text.

### C.1.3 Constructing test set

*Why 6 nodes?* Considering the maximal number of nodes is 7, 6 may seem "too close". On the other hand, picking a graph and excluding its subtree means that this subgraph will *never* appear in the training set. This is quite aggressive, for example creating a test set of 10% of the data only requires 35 such graphs whose subtrees are excluded. These graphs have an average of $272 \pm 205$ descendants. Choosing graphs of 5 nodes would exclude an average of $\sim 4700$ graphs per pick, meaning that a test set of 10% of the data would only stem from the exclusion of 2 or 3 graphs. This may not have the diversity we desire, thus our choice of 6 nodes.

Finally, we believe this is an interesting choice that relates to real-world applications of GFlowNets. In drug-discovery of small molecules, given the enormous size of the state space, it is likely for most subgraphs of a sufficient size to never appear in the training set (because a model can only train for so many iterations in practice). In a graph generation context we are thus interested in how a model generalizes to subgraphs it's never seen. In this proposed benchmark, 67994 of the 72296 states are graphs of 7 nodes; only excluding graphs of 6 or 7 nodes thus covers most of state space.

## C.2 Details for hypergrid and sequence environments and tasks

### C.2.1 ($M \times M$) Hypergrid:

Grid environments have been used in GFlowNet papers since their inception (Bengio et al., 2021) as a sanity check environment, presumably inheriting this custom from Reinforcement Learning. We continue the tradition and report results on a 2D grid environment. In it, the agent has 3 actions, move in the $x+$ direction, move in the $y+$ direction, or stop. The agent navigates on a grid of size $M$ (64 in our experiments), and is forced to stop if any of the coordinates reaches $M - 1$.

As a reward signal we use functions used in past work (Bengio et al., 2021; Jain et al., 2023).

### C.2.2 Bit Sequences:

Auto-regressive sequence environments have been used in a variety of past GFlowNet work (Malkin et al., 2022; Madan et al., 2022; Pan et al., 2023). In this work, we consider a bit sequence environment akin to that originally introduced by Malkin et al. (2022). We consider a reward of the form $R(x) = \exp(\frac{-\min_{m \in \mathcal{M}} d(x,m))}{l} \times 10)$, where $d$ is the Levenshtein edit distance between an input sequence $x$ and the closest "mode" sequence $m$, $\mathcal{M} \subset \mathcal{X}$ is the set of mode-sequences, and $l$ is the max sequence length. We select $|\mathcal{M}| = 60$ sequences from $\mathcal{M}$ uniformly at random given the set $\mathcal{X}$ of all possible bit sequences up to length $l = 15$ to be the "mode" sequences (we do this once at the start of each run). This yields a discrete state space of $65,535$ states, comparable in size to our graph generation environment.

## C.3 Monotonic reward transformation

We consider a monotonic transform of the form $\mathcal{H}_\gamma(\log R(x)) = e^{-\gamma \log R(x)} \log R(x)$. As described earlier, the transform $\mathcal{H}_\gamma$ maps $\log R'(s) = \mathcal{H}_\gamma(\log R(s)), \forall s \in \mathcal{S}$ such that $\log R'(s)$ is skewed towards higher reward states as a function of some parameter $\gamma$ (see §C.3 for further details. Note, $\mathcal{H}_\gamma$ is monotonic for values of $\log R(x) \leq 0$ and $\gamma \geq 0$. In our graph and sequence generations tasks, $\log R(x) \leq 0, \forall x \in \mathcal{X}$, hence we are able to use $\mathcal{H}_\gamma$ as a monotonic transform for $\log R(x)$. We show an example of $\mathcal{H}_\gamma$ applied to the cliques task in Figure 8 for various value sof $\gamma$.

## C.4 Evaluation metrics

To measure the generative modeling performance of GFlowNets we consider the Jensen-Shannon (JS) divergence and the MAE between the learned $\log p(x; \theta)$ and the true $\log p(x)$. For JS divergence, we compute:

$$\mathrm{JS}(p(x), p(x; \theta)) = \frac{1}{2}\mathrm{KL}(p(x)\|Q) + \frac{1}{2}\mathrm{KL}(p(x; \theta)\|Q), \tag{3}$$

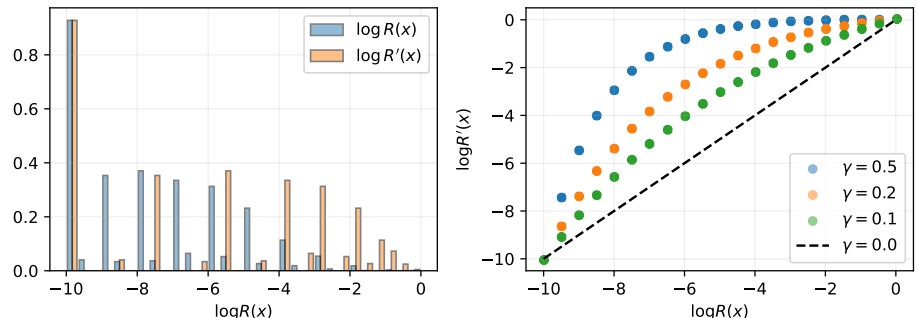

Figure 8: Example of monotonic transform $\mathcal{H}_\gamma$. (left) distributions of $\log R(x)$ and $\log R'(x)$ for true log-rewards and skewed log-rewards (using $\gamma = 0.2$). (right) plot of $\log R(x)$ versus $\log R'(x)$ for various values of $\gamma$. Points above the $\gamma = 0$ line yield skew towards higher log-reward values.

where $Q = \frac{1}{2}(p(x) + p(x; \theta))$. For the MAE, we simply take the absolute error $|\log p(x; \theta) - \log p(x)|$ and average over the cardinality of the state space.

### C.5  Model architectures

For all graph experiments we use a modified graph transformer (Veličković et al., 2017; Shi et al., 2020). On top of the normal attention mechanism, we augment the input of each layer with the output of one round of message passing (using layers from the work of Li et al., 2020) with a `sum` aggregation–we found that this was useful in tasks where counting was required. We use 8 layers with 128-dimensional embeddings and 4 attention heads; we use this architecture after having tried different numbers of layers and embeddings in order to validate our task; this is coincidentally represented in Fig. 1(a).

For sequence tasks, we use a vanilla transformer (Vaswani et al., 2017) with 4 layers of 64 embeddings and 2 attention heads. We did not search for hyperparameters, since this setup has been used in prior work (Malkin et al., 2022).

For grid tasks we use a LeakyReLU MLP with 3 layers of 128 units. The input is a one hot representation of each coordinate.

We use the Adam optimizer with learning rate 0.0001 for all models.

### C.6  Implementation details

Our experiments are implemented in Pytorch and Pytorch Geometric. All experiments were run on a HPC cluster of NVIDIA A100 100GB GPUs for a total of approximately 2000 GPU hours. Only 1 GPU is required for Eeach individual seed run of an experiment, typically taking between 24 hours to 3 days to complete, depending on the experiment. Throughout the conception of this work, there were a small set of preliminary and failed experimental runs as ideas were being formed.

## D  Additional Graph Task Experiments

### D.1  Graph neural network architectures for GFlowNet graph task experiments

We tried a few GNN architectures. GATs appeared the best. While they are not the simplest GNNs, they may be the most general (because of the attention mechanism). We compared GATs, GCNs and GINs the experiment of Figure 2 (i.e. simply training GFlowNets online on-policy)s. We were careful to ensure that all models have almost-equal numbers of parameters. We find GATs to be superior, but we acknowledge that this may not be the case in tasks beyond this benchmark, and that we have obviously spent more time tuning our GAT model than these. We believe it unlikely that the choice of model would affect our observations.

## D.2 Other GFlowNet objectives

We elected to consider SubTB(1) since it has been shown to yield improved results over TB and Detailed Balance (DB). Furthermore, we consider the Trajectory Balance (TB) and Flow Matching (FM) objectives in a set of offline experiments to assess how GFlowNets assign probability mass to unvisited areas of the state space (see D.3). In Table 5, we show that SubTB(1) performs best (compared to offline TB and offline FM) when ranking states in the graph tasks.

## D.3 GFlowNets learn rank preservation of unseen states

Table 5: Comparing the rankings given by different ways of training a model on a dataset. Standard deviations are over 4 runs. In the last three rows the number in brackets is the number of optimal objects in the test set.

| Task | **Supervised** | **Distilled** $F$ | **Distilled** $P_F$ | **offline** TB | **offline** subTB | **offline** FM |
|---|---|---|---|---|---|---|
| | \multicolumn{6}{c}{Test set Spearman correlation} | | | | | |
| **cliques** | $0.92 \pm 0.02$ | $0.92 \pm 0.01$ | $\mathbf{0.93} \pm \mathbf{0.00}$ | $0.88 \pm 0.01$ | $0.90 \pm 0.01$ | $0.86 \pm 0.03$ |
| **neighbors** | $\mathbf{0.98} \pm \mathbf{0.00}$ | $\mathbf{0.98} \pm \mathbf{0.00}$ | $\mathbf{0.98} \pm \mathbf{0.00}$ | $0.96 \pm 0.00$ | $0.96 \pm 0.00$ | $0.96 \pm 0.00$ |
| **count** | $\mathbf{0.98} \pm \mathbf{0.00}$ | $\mathbf{0.98} \pm \mathbf{0.00}$ | $\mathbf{0.98} \pm \mathbf{0.00}$ | $0.95 \pm 0.01$ | $0.95 \pm 0.00$ | $0.96 \pm 0.00$ |
| | \multicolumn{6}{c}{Test set top-100 Spearman correlation} | | | | | |
| **cliques** | $0.66 \pm 0.02$ | $0.74 \pm 0.10$ | $\mathbf{0.83} \pm \mathbf{0.04}$ | $0.67 \pm 0.10$ | $0.69 \pm 0.11$ | $0.53 \pm 0.13$ |
| **neighbors** | $\mathbf{0.87} \pm \mathbf{0.00}$ | $\mathbf{0.87} \pm \mathbf{0.00}$ | $\mathbf{0.87} \pm \mathbf{0.00}$ | $0.86 \pm 0.01$ | $0.86 \pm 0.00$ | $0.86 \pm 0.01$ |
| **count** | $\mathbf{0.44} \pm \mathbf{0.00}$ | $\mathbf{0.44} \pm \mathbf{0.00}$ | $\mathbf{0.44} \pm \mathbf{0.00}$ | $0.24 \pm 0.06$ | $0.24 \pm 0.06$ | $0.37 \pm 0.02$ |
| | \multicolumn{6}{c}{Avg Rank of optimal objects} | | | | | |
| **cliques** *(15)* | $21.5 \pm 14.5$ | $39.3 \pm 30.2$ | $\mathbf{13.3} \pm \mathbf{5.3}$ | $18.1 \pm 4.9$ | $\mathbf{12.1} \pm \mathbf{2.0}$ | $\mathbf{17.7} \pm \mathbf{5.2}$ |
| **neighbors** *(48)* | $\mathbf{23.5} \pm \mathbf{0.0}$ | $\mathbf{23.5} \pm \mathbf{0.0}$ | $\mathbf{23.5} \pm \mathbf{0.0}$ | $43.4 \pm 5.5$ | $43.6 \pm 9.4$ | $40.7 \pm 2.2$ |
| **count** *(7)* | $\mathbf{3.0} \pm \mathbf{0.0}$ | $\mathbf{3.0} \pm \mathbf{0.0}$ | $\mathbf{3.0} \pm \mathbf{0.0}$ | $62.7 \pm 12.3$ | $61.1 \pm 15.1$ | $25.6 \pm 4.2$ |

We want to assess more precisely where GFNs put probability mass, in particular in states they've never visited. More specifically though, GFNs are often used to find *likely* hypotheses, i.e. generate objects "close enough" to the argmax(es).

Consider the following scenario, which is a common way to use GFNs. Some dataset of objects and scores $\mathcal{D} = \{(x_i, y_i)\}$ is given to us. We train a reward proxy to regress to $R(x_i; \theta) = y_i$, and then train a GFN on $R(x; \theta)$ in order to generate $x$s (and commonly, find the most "interesting" $x$). If we assume that supervised learning is "as good as it gets" to approximate $R$, then the task of finding $\arg\max_x R(x)$ by finding $\arg\max_x R(x; \theta)$ should also be as good as it gets. This prompts us to ask, how close are GFNs to this ideal?

In the following experiment, we look at three measures. First, the Spearman rank correlation on the probabilities $p(x; \theta)$ (or $R(x; \theta)$ for the supervised model) of the test set, second we do the same for the top-100 graphs in the test set, and finally we look at the mean rank of the set of optimal $x$s in the test set (e.g. if there are 7 optimal graphs in the test set and the model perfectly predicts their rank to be $[0, 1, 2, 3, 4, 5, 6]$ then the mean rank is $(0 + 1 + 2.. + 6)/7 = 3.0$).

These measures are intended to be proxies for how likely it is that a model trained on some dataset would be able to generate objects close enough to the optimal objects.

This third measure is also inspired by the following fact: it has been observed in a few setups that even though trajectory balance objectives appeared to fit the distribution much better than flow matching, flow matching can be more efficient at producing high-reward samples (which seems counter-intuitive).

We thus compare distilling edge flows to policies, as well as TB objectives to the FM objective. Specifically for GFN objectives (TB, subTB, FM), we use an offline paradigm where we sample trajectories by sampling an $x$ uniformly at random from $\mathcal{D}$ and going backwards with $P_B$ uniformly at random (this is presumably a somewhat ideal training condition).

We report the results in Table 5. The results are quite dense, so let us make a few observations:

- When considering the entire test set, the supervised model is indeed the gold standard. Offline GFN models are slightly worse. This isn't too surprising, and is in fact reminiscent of results related to the difficulty of training value functions via TD in RL (Bengio et al., 2020).

- When considering either the top-100 or the optimal objects, we see surprising results for the cliques task (the hardest task)
  - for the top-100 objects, the distilled models do better, and the offline TB models are on-par with (and occasionally better than) the supervised model
  - for the rank of optimal objects, not only does distilled $P_F$ do better, the offline GFN methods do better as well.

  Recall that some test set information is leaking into training; it may explain the advantage of distilled models, but not the advantage of offline GFN models.

- TB and subTB are generally better at fitting *ranking* than FM, *except* when it comes to optimal objects. This reproduces past observations, suggesting flow matching over-weights high-reward states, even outside of its training data; we unfortunately do not have a coherent explanation:
  - $F(s, s')$ distillation being worse than $P_F$ distillation means modeling $F$ is probably not the advantage;
  - This phenomenon is consistent across training, although not on every task.
  - This phenomenon is amplified when *not* correcting for idempotent actions. This is expected (Ma et al., 2023), but useful to measure.

We repeat these three measures with different train-test ratios, shown in Figure 17, where we see that the results are as expected, and consistent with Table 5. To summarize: *GFlowNets implicitly learn to put more mass on high-reward objects than low-reward objects outside their training set.*

The above is interesting, but is still just a statement about *ranking* between unseen states. Obviously this says nothing about the total probability mass given to unseen states.

### D.4 Distilled flow experiments over only unseen states

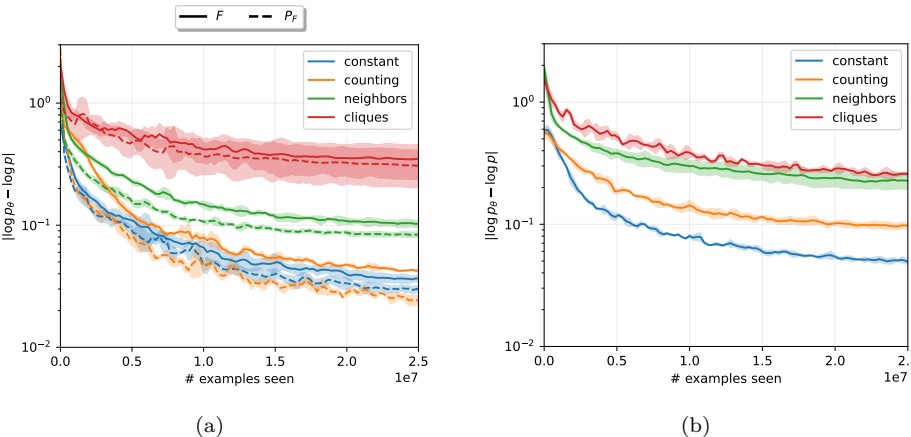

(a)                                   (b)

Figure 9: MAE error for only unseen states on constant, counting, neighbors, and clique graph generations tasks. (a) evaluation for model trained to distill edge flows and policies. (b) evaluation an online trained GFlowNet via SubTB(1). We see that considering only the unseen states for MAE metric leads to comparable observation to the case of evaluating MAE over all states. This is expected since error on the unseen states should be driving the model's overall evaluation performance when evaluating on the entire state space. Note, for the online GFlowNet, there is no guarantee that the model has not "seen" the left out unseen states in this evaluation.

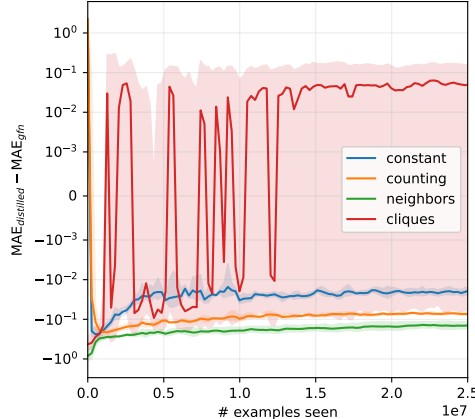

Figure 10: MAE gap between online trained GFlowNet via SubTB(1) and distillation trained model regressing to $P_F$ on only unseen states. We observe that when considering only unseen states, results are consistent with evaluation over the entire state space. This is also expected since error on the unseen states should be driving the model's overall evaluation performance when evaluating on the entire state space.

### D.5 Additional memorization experiments

In Figure 11 we look at reward corruption. This is another setting considered by Zhang et al. (2017; 2021) to achieve de-structuring of the form $s \perp\!\!\!\perp \tilde{R}(s)$.

In Figure 12 we look at online trained vs distilled GFlowNets with true vs shuffled rewards.

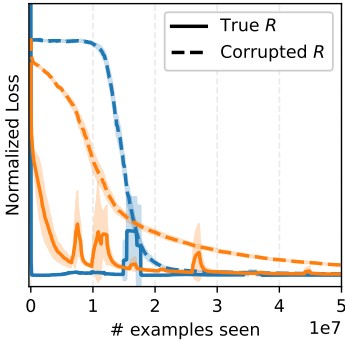

Figure 11: Memorization gap training curves for the constant reward on the graph generation task. In this experiment, we consider the case where Gaussian noise corrupts the constant reward signal of the form $\tilde{R}(x) = R(x) + \epsilon, \epsilon \sim \mathcal{N}(0, \sigma)$, thus inducing de-structuring. Here we consider $\sigma = 2$. We observe behaviour consistent with that found in §4.3.

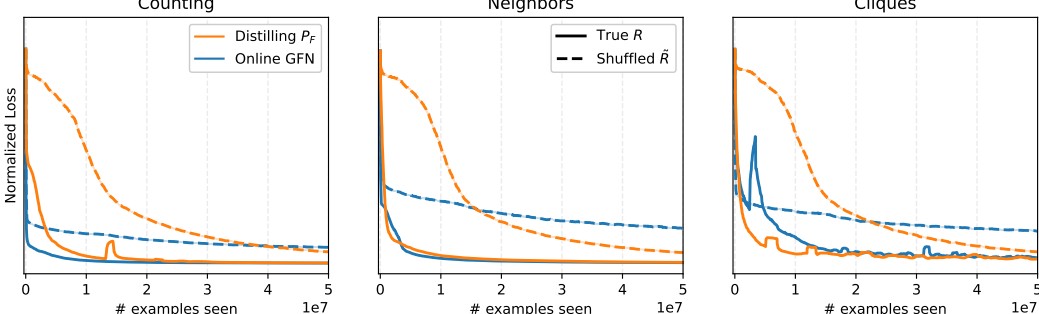

Figure 12: Memorization gap training curves for counting, neighbors, and cliques tasks for distilled (regressing to $P_F$) and online trained GFlowNet. We observe results are consistent with findings in §4.3 for the online trained GFlowNet.

### D.6 Full Experimental results for offline and off-policy training of GFlowNets

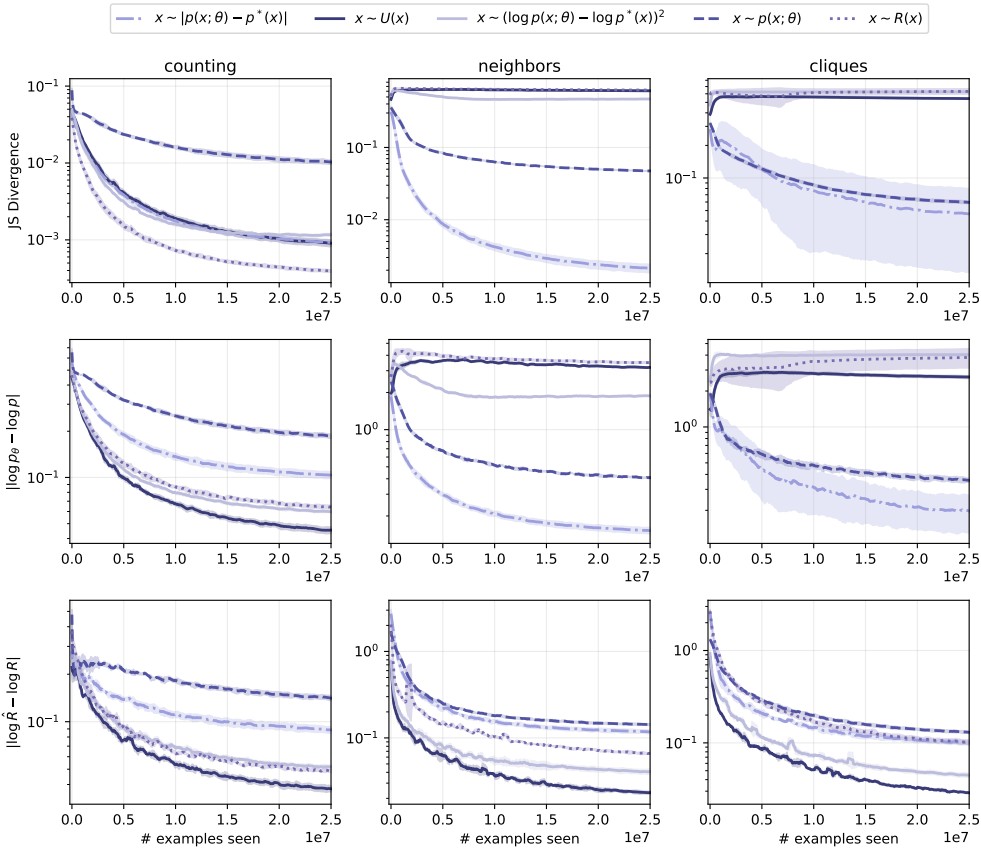

Figure 13: Evaluation curves for offline and off-policy trained GFlowNets on counting, neighbors, and cliques graph generations tasks for different choices of $\mathbb{P}_{\mathcal{X}}$ when training using the full dataset (no test set). Model performance is dependent on the choice of $\mathbb{P}_{\mathcal{X}}$. When considering evaluation on the JS divergence and MAE metrics, depending on task and some choices of $\mathbb{P}_{\mathcal{X}}$, $p(x)$ is not adequately approximated. However, regardless of task, offline and off-policy trained GFlowNets appear to robustly learn $R$.

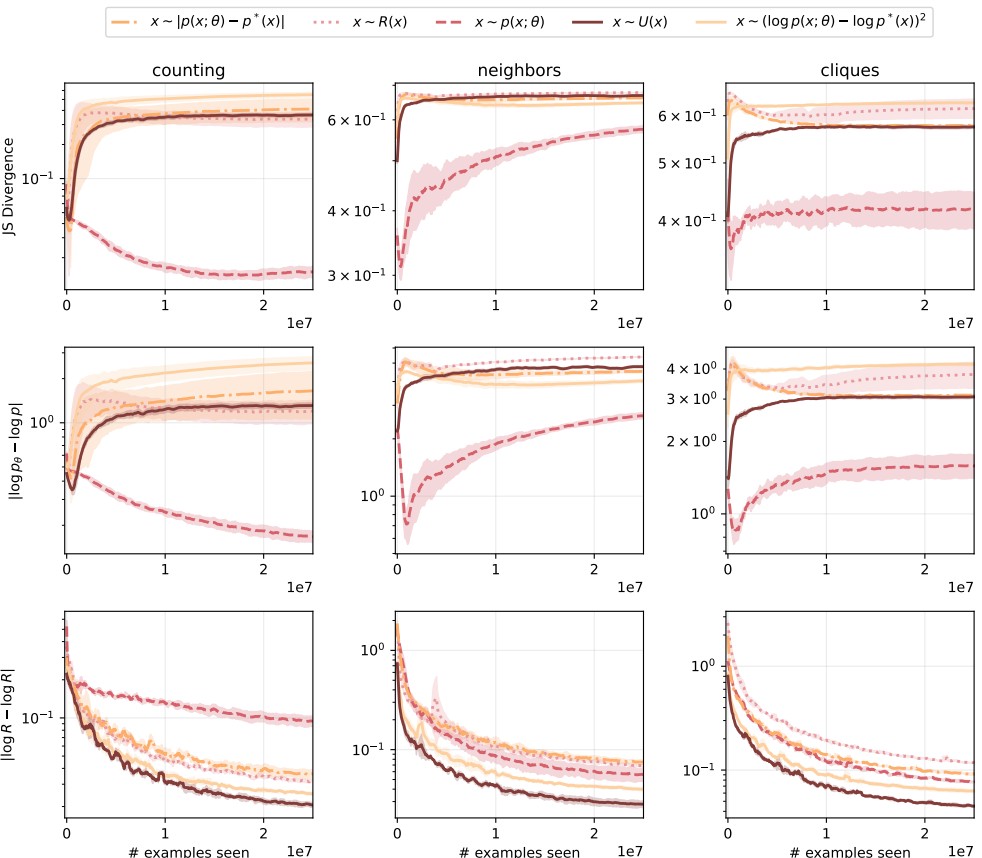

Figure 14: Evaluation curves for offline and off-policy trained GFlowNets on counting, neighbors, and cliques graph generations tasks for different choices of $\mathbb{P}_{\mathcal{X}}$ when training using a subset of the full dataset (90%-10% train-test split). In this setting, agnostic to the choice of $\mathbb{P}_{\mathcal{X}}$ and graph generation task, we observe the GFlowNet models struggles to converge when considering JS divergence and MAE distributional metrics for approximating $p(x)$. However, we observe the GFlowNet models remains robust when learning $R$.

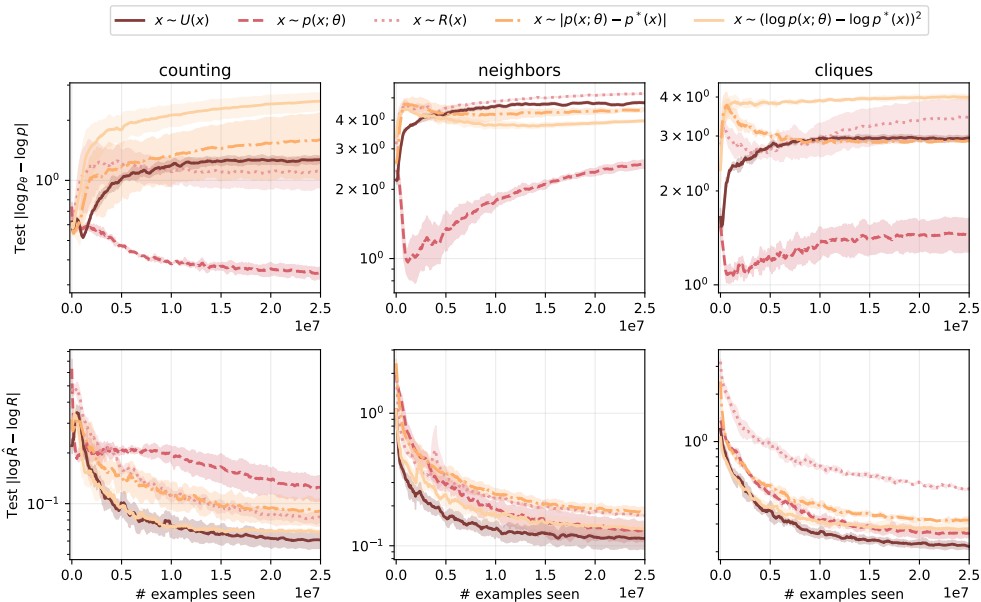

Figure 15: Continuation of Figure 14, but evaluated over only the unseen sates (test states). We observe performance is consistent with the evaluations over the entire state space (Figure 14).

## D.7    Offline GFlowNets inadequately assign probability mass

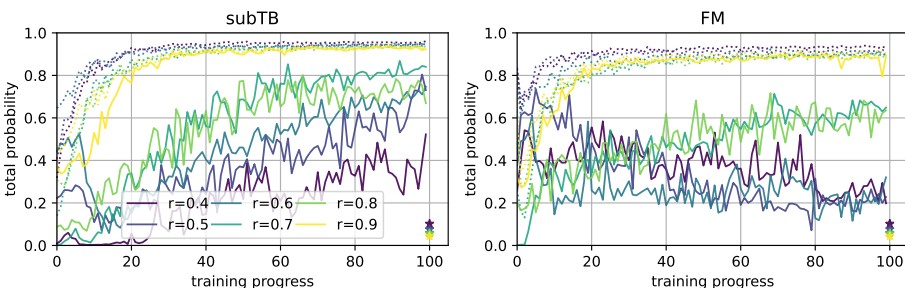

Figure 16: The total probability of the test set (dotted) and of the top-100 objects within the test set (full lines); means of 4 seeds. Each color is a different training/test split ratio. As the model has access to more data, it seems to overfit *more*; it gives more total probability to very few states, but particularly, to high reward states. We also plot in the bottom right (stars) what the "true" total probability of the top-100 objects should be (it varies because of the random seeds and test set size, we show the seed average).

Let's dig into the experiment of §D.3. We've established that offline GFNs manage to rank unseen states fairly well, but this says nothing about how much probability it gives those states, only that the relative probabilities are well behaved.

In Figure 16 we show how much *total probability* (i.e. $\sum_{i \in \text{test}} p(x_i; \theta)$) is allocated to the test set. In particular, we show the total probability of the whole test set, as well as of the top-100 objects within the test set.

We believe the following result is unexpected: **as the training set grows larger, high-reward unseen objects take up more total probability**.

This seems to be counter to our intuition on generalization; which suggests that as a model gets more training data, it should extrapolate better (which for a probability model should mean that $p(x; \theta)$ should get closer to $p(x)$).

This result seems to be both terrible news and great news. First the bad news; this result suggests that naively applying GFN objectives to an offline training regime simply doesn't work. Indeed, intuitively because $Z$ is a free variable in GFNs, one of two things seems likely to happen: either the model ends up learning $Z = \sum_{i \in \text{train}} R(x_i)$ and giving 0 probability to the test set, or the model ends up learning an arbitrary large $Z$ and giving 0 probability to the training set. The good news; while the latter seems to happen (the training set ends up with fairly low total probability), there seems to be *structure* to which states receive the most probability. This again seems particularly useful in the scenario where we are looking for the most interesting (high-reward) $x$.

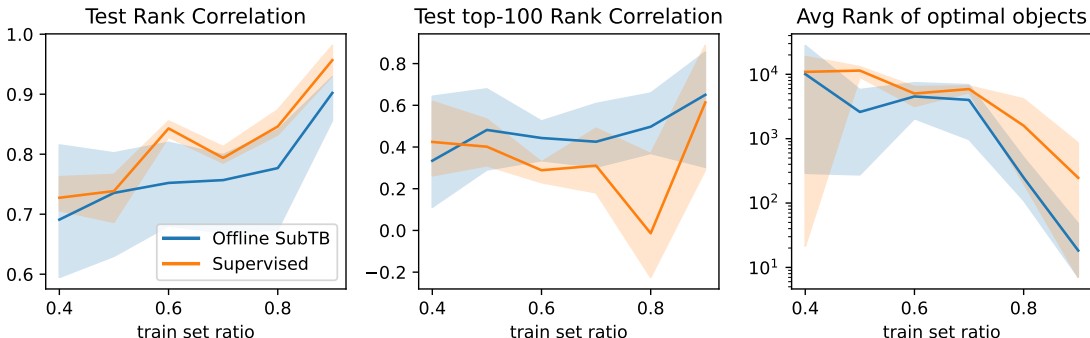

Figure 17: Rank-related metrics during offline GFN training and supervised regression as a function of the size of the training set. Averages are over 4 seeds (which influence the construction of the test).

# E   Sequence and Hypergrid Experiments

## E.1   Online and distilled training

### E.1.1   Online trained models

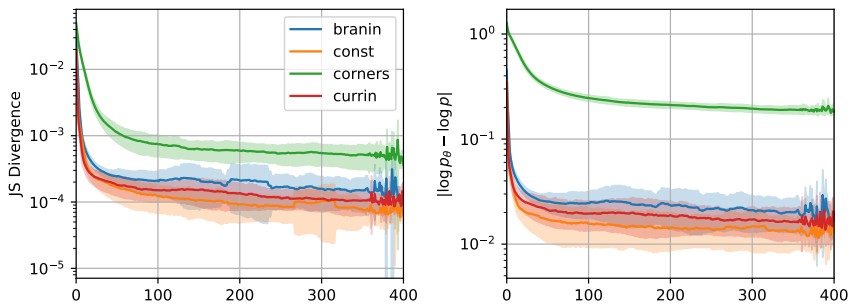

Figure 18: Training a GFlowNet (online and on-policy) on 4 different **hypergrid** tasks. Corners, the most frequently used task in the hypergrid environment, appears to be most difficult for learning $p(x)$.

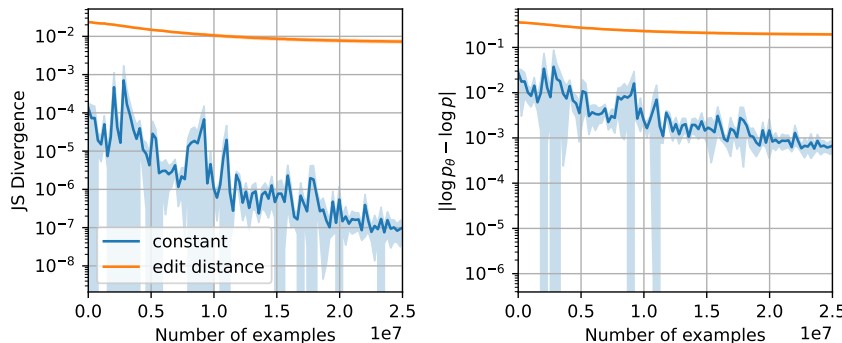

Figure 19: Training a GFlowNet (online and on-policy) on 2 different **sequence** tasks. The edit distance task appears to be significantly more difficult than the constant task for learning $p(x)$. Given the auto-regressive nature of the sequence environment, it is not unsurprising how well a GFlowNet performs on the constant task (i.e. the sequence environment is not challenging on its own). In contrast, we reify that the edit distance reward is indeed a challenging task in this environment.

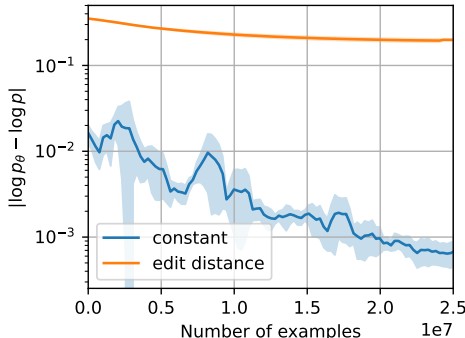

Figure 20: MAE error for only test states on the sequence task for online and on-policy. Note that for this experiment, the test states are not necessarily unvisited by the model

### E.1.2 Distilled flows and forward policy models

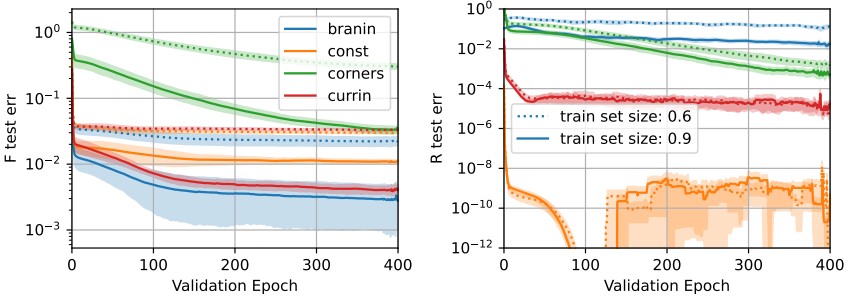

Figure 21: Training a MLP on 4 different tasks to regress to $F$ (left) and $R$ (right) in the **hypergrid** environment. Task difficulty between supervised models and online and on-policy trained GFlowNets appears consistent.

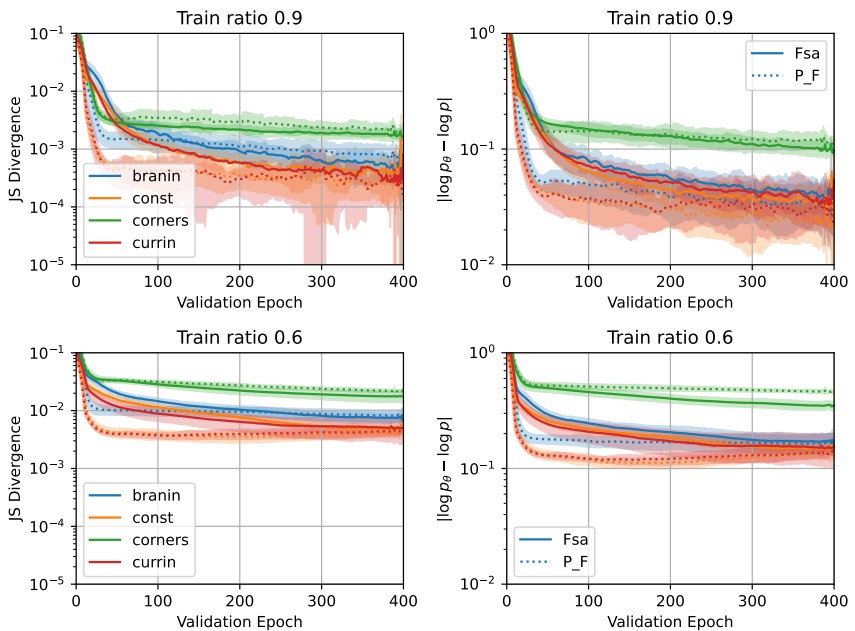

Figure 22: Training a model to distill edge flows and policies on 4 different **hypergrid** tasks for 90%-10% train-test split (top) and 60%-40% train-test split (bottom). It generally seems that (a) doing so recovers the intended distribution fairly well, and (b) modeling $P_F$ appears easier than modeling $F$, insofar as it recovers $p(x)$ better, in the hypergrid environment.

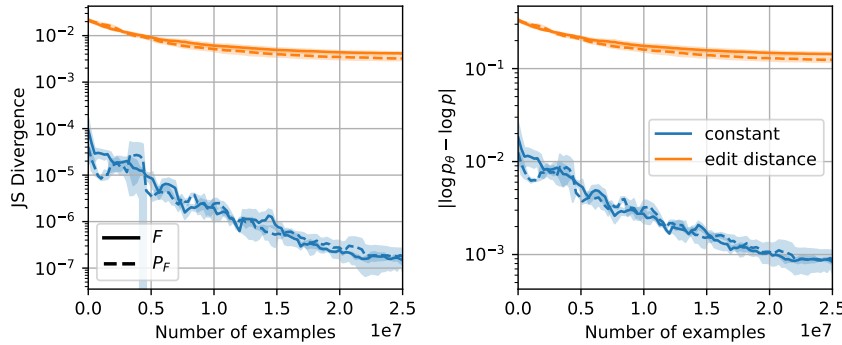

Figure 23: Training a model to distill edge flows and policies on 2 different **sequence** tasks for 90%-10% train-test split. It generally seems that (a) doing so recovers the intended distribution fairly well, and (b) modeling $P_F$ appears easier than modeling $F$, insofar as it recovers $p(x)$ better, in the sequence environment for the edit distance reward. For the constant reward, there appears to be no apparent difference in difficulty between learning $P_F$ or $F$.

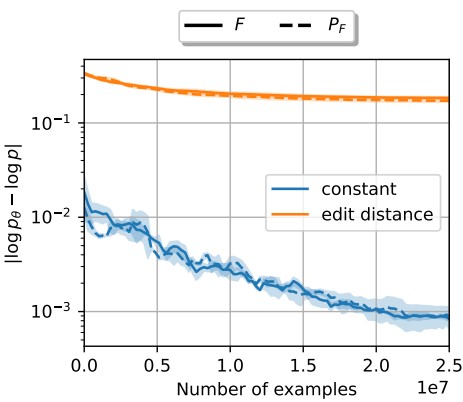

Figure 24: MAE error for only test states on the **sequence** task for distilled model, regressed to $P_F$. Note that for this experiment, the test states are unvisited by the model.

## E.2 Reward transformation

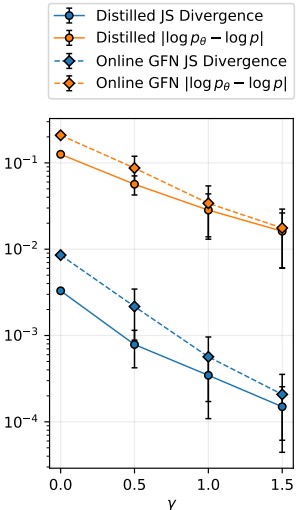

Figure 25: Training models distilled to $P_F$ and GFlowNet models trained online/on-policy for a range of monotonic skew values $\gamma$ on the **sequence** task. Transforming the distribution of the edit distance reward to contain a larger proportion of high reward values generally improves the performance for approximating $p(x)$.

## E.3 Memorization gap experiments

### E.3.1 Memorization in hypergrids

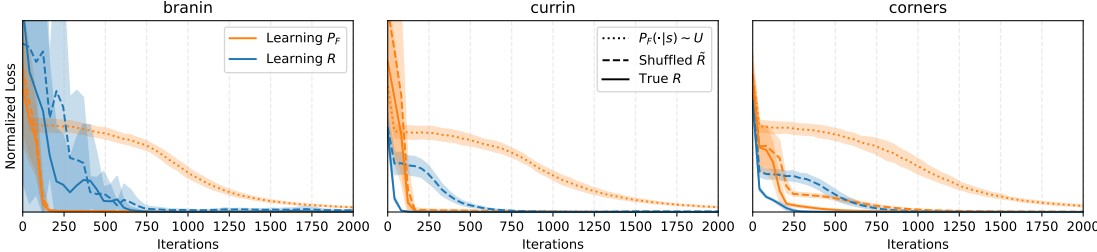

Figure 26: Memorization gap training curves for branin, currin, and corners tasks on the **hypergrid** environment. Maintaining *flow* structure in the learning problem (learning $P_F$ under shuffled $\tilde{R}$) generally induces a smaller *memorization gap* relative to the fully de-structured setting. Note that for the branin reward, there is not apparent difference between learning the true $R$ and the shuffled $\tilde{R}$, suggesting the reward alongside the hypergrid environment induces a task with minimal difficulty.

### E.3.2 Memorization in sequences

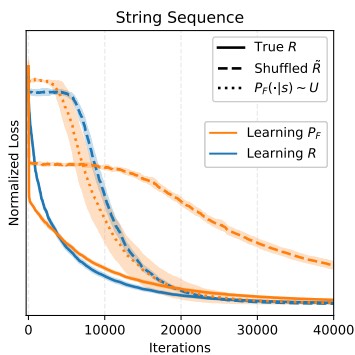

Figure 27: Memorization gap training curves for the edit distance task on the **sequence** environment. Maintaining *flow* structure in the learning problem (learning $P_F$ under shuffled $\tilde{R}$) generally induces a smaller *memorization gap* relative to the fully de-structured setting.

### E.4 Offline and off-policy training

### E.4.1 Offline and off-policy in hypergrids

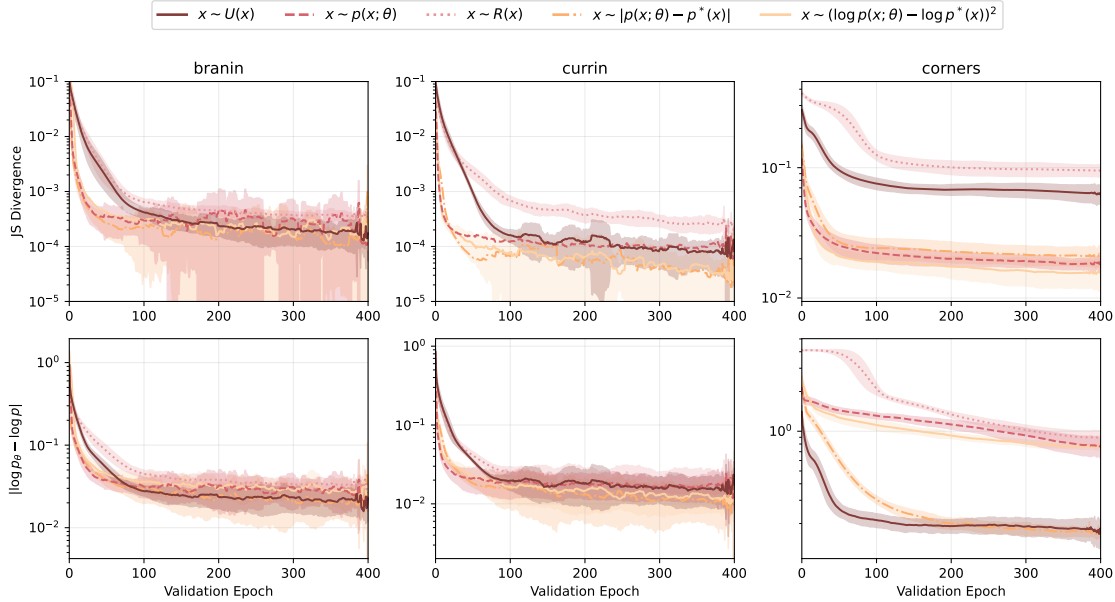

Figure 28: Evaluation curves for offline and off-policy trained GFlowNets on branin, currin, and corners **hypergrid** tasks for different choices of $\mathbb{P}_{\mathcal{X}}$ when training using a subset of the full dataset (90%-10% train-test split). In this setting, agnostic to the choice of $\mathbb{P}_{\mathcal{X}}$ and hypergrid reward, we observe the GFlowNet models converge when considering both the JS divergence and the MAE distributional metrics for approximating $p(x)$. This differs from the graph generation tasks where models struggle to converge on these metrics in this setting. For the more difficult task (corners), sampling $x$ from sets that include the proxy-policy $p(x; \theta)$ result in the best performance on JS divergence while sampling $x \sim \mathcal{U}$ and $x \sim |\log p(x; \theta) - \log p(x)|$ results in the best performance on the MAE metric.

### E.4.2 Offline and off-policy in sequences

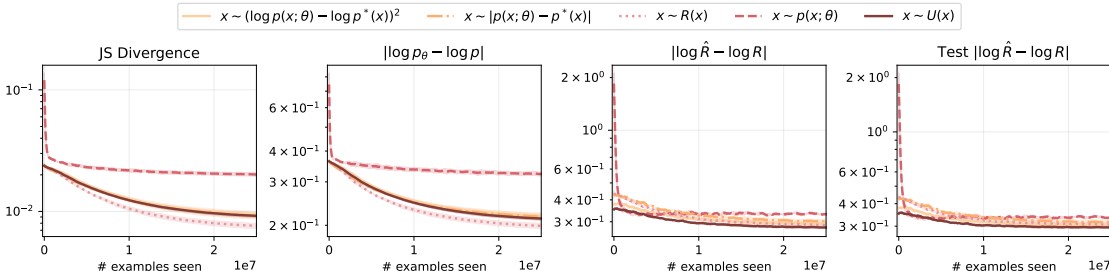

Figure 29: Evaluation curves for offline and off-policy trained GFlowNets on edit distance **sequence** task for different choices of $\mathbb{P}_{\mathcal{X}}$ when training using a subset of the full dataset (90%-10% train-test split). In this setting, agnostic to the choice of $\mathbb{P}_{\mathcal{X}}$, we observe the GFlowNet models converge when considering JS divergence and MAE distributional metrics for approximating $p(x)$. This differs from the graph generation tasks where models struggle to converge on these metrics in this setting.

