# OpenReview forum: "Investigating Generalization Behaviours of Generative Flow Networks"
_TMLR — Accepted by TMLR_

### Review · Reviewer_ePyU · 2024-12-25

**Summary Of Contributions:**

- This paper investigates generalization properties of GFlowNets using a new synthetic environment.
 - Three hypotheses are made about these generalization properties:
    - GFlowNets generalize under a narrow set of distributions.
    - GFlowNets generalize by learning structure.
    - Generalization performance is driven mostly by reward complexity rather than distribution properties such as temperature and skewness.
 - Three core experimental settings are used to test these hypotheses:
    - Distillation - using the tractability of the new synthetic dataset, GFlowNets can be trained and tested while controlling for non-ideal factors of the GFlowNet training setup.
    - Memorization - various methods of ablation structure from the data / model are used to probe the generalization / memorization properties of GFlowNets.
    - Offline and off policy training - generalization performance can be measured as a function of differing training distributions.
 - The proposed synthetic environment is based on a graph building problem with three levels of difficulty.
    - This task is more difficult that conventional synthetic tasks: hypergrid and sequence tasks.
    - Although the task is more difficult, important quantities for benchmarking (e.g., exact likelihoods, flows) are tractable.

**Audience:**

Yes

**Claims And Evidence:**

Yes

**Requested Changes:**

My requested changes are mostly minor, as specified in the weaknesses section.
I also recommend correcting the typo: "asses" -> "assess"

**Strengths And Weaknesses:**

__Strengths__:
 - The paper is generally written very well and most points are communicated clearly and concisely.
 - The relative difficulty of the three problems defined in the graph environment are verified empirically and are sound.
 - Experiments are designed well to probe specific properties of GFlowNets.
 - Experiments retest and challenge conventional beliefs about GFlowNets on the new and more difficult task.
 - Interpretation of results is reasonable without extraordinary claims, and clearly identifies important points where some results are not completely conclusive but are still interesting.
 - The proposed synthetic environment and the results from the experiments are significant and are valuable for building our understanding of GFlowNets.
 - Many of the details are provided in the appendix and referenced appropriately.

__Weaknesses__:
 - In the introduction, it is not immediately clear how experiment 1 supports hypotheses 2 and 3. It seems the experiment 1 does not directly support hypothesis 2, but is used as a stepping stone to support the hypothesis by using the distillation method in experiment 2. Distillation alone seems to only test generalization in general, but without considering experiment 2, it is not clear how it is testing the learning of structure. It is not revealed until the experiments how Hypothesis 3 is supported by the distillation (via monotonic transformation of the reward).
 - The explanation of experiment 2 (memorization in section 3.2) could be made more clear within the section. Between sections 3.1 and 3.2 there is a slight change in language where it is potentially not clear that the true flows are computed from the shuffled rewards. This point is actually made clear at the end of section A of the appendix on the definition of "True flows", which could at least be referenced within section 3.2 or clarified with an additional sentence.
 - In section 3.2 is the shuffling of rewards is amongst all states or just the terminal states?
 - \tilde{P}_F(s'|s) in table 3 is not defined anywhere but can be assumed to be policy logits derived from the shuffled rewards.

---

> ### Author Response · Authors · 2025-02-18
>
> We thank the reviewer for their constructive comments and insightful questions. We are excited to see that the reviewer found our paper "very well written" with "concise and clear" exposition and that our "experiments are designed well to probe specific properties of GFNs". We are also happy to see that the reviewer found our benchmark environment and corresponding findings from our empirical study as "significant and valuable for building our understanding of GFNs". In the following, we provide detailed responses to the reviewer's questions.
>
> >In the introduction, it is not immediately clear how experiment 1 supports hypotheses 2 and 3. It seems the experiment 1 does not directly support hypothesis 2, but is used as a stepping stone to support the hypothesis by using the distillation method in experiment 2. Distillation alone seems to only test generalization in general, but without considering experiment 2, it is not clear how it is testing the learning of structure. It is not revealed until the experiments how Hypothesis 3 is supported by the distillation (via monotonic transformation of the reward).
>
> Thank you for pointing this out. You are correct in that experiment 1 is used as a stepping stone towards setting up experiment 2, where experiment 2 serves as the means for probing hypothesis 2. Similarly, the setup of experiment 1 is, in part, also used to test hypothesis 3. We will adjust the text in the introduction to clarify these points.
>
> >The explanation of experiment 2 (memorization in section 3.2) could be made more clear within the section. Between sections 3.1 and 3.2 there is a slight change in language where it is potentially not clear that the true flows are computed from the shuffled rewards. This point is actually made clear at the end of section A of the appendix on the definition of "True flows", which could at least be referenced within section 3.2 or clarified with an additional sentence.
>
> Thank you for this suggestion. We agree that referencing section A on the definition of "true flows" and explicitly stating that they are computed from the shuffled rewards in section 3.2 would help clarify the explanation of experiment 2 (memorization gap experiments). We will adjust the text accordingly to improve clarity.
>
> >In section 3.2 is the shuffling of rewards is amongst all states or just the terminal states?
>
> The shuffling of rewards is done across all states. Since the state space is of reasonable size (s.t. we can tractably compute $P_F(s'|s)$ exactly), we are also able to tractably shuffle and compute the corresponding $\tilde{P}_F(s'|s)$.
>
> >\tilde{P}_F(s'|s) in table 3 is not defined anywhere but can be assumed to be policy logits derived from the shuffled rewards.
>
> Yes, $\tilde{P}_F(s'|s)$ is the policy logits derived from the shuffled rewards. We will add this to the body of the text.
>
> We once again thank the reviewer for their time and effort in reviewing our work. We believe that through this rebuttal and with the helpful comments from the reviewer, the quality of our paper will improve. We are more than happy to answer any additional questions that may arise.

---

### Review · Reviewer_3T6g · 2025-01-04

**Summary Of Contributions:**

The paper presents a synthetic task for studying the generalization properties of Gflownets (GFNs). The task consists of learning graph edits on graphs of up to 7 nodes, such that the resulting graph distribution follows the reward distribution induced by 3 + 1 debugging tasks

1. Clique(G)=n_CC(G)-n_C(G)+n(G)

    where n_CC is the number of 4 cliques in which 3 or 4 nodes have the same color (max number of 4 cliques is 35, max number of clqiues is 120, for reference), then take exp => clipped to have logR be between -10 and up to 0.

2. count the number of nodes with an even number of neighbours of the opposite color

3. counting is number of red and green and assigns ground truths

4. (debugging) constant assignment of uniform reward everywhere


Given the chosen graph state space yields a size of ~72k states, the authors can then derive ground truth state probabilities and flow transitions from the tasks, and perform a series of studies to investigate a set of key hypotheses around GFNs ( generalization under a narrow set of distributions, based on structure, driven by the complexity of the reward as distributed across the dynamics, not the distributional properties of the reward over end states).

**Audience:**

Yes

**Broader Impact Concerns:**

theoretical study which does not require ethical implications, beyond the 2k GPU hours which we all inflict onto our planet with each paper

**Claims And Evidence:**

Yes

**Requested Changes:**

None of these are critical, but I do think they'd make the paper _even better_:


Suggestions:

1. the clarification with making definitions unambiguously attributable

2. if computational power applies, one could validate the GAT strength as follows:

    1. assign a unique label (e.g. in a decimal one hot encoded way) to each graph and verify the model can learn this to zero training loss (meaning it can learn to identify each graph uniquely)

    2. verify the model can learn to regress the dynamic programming value function for each state to zero training loss


    neither of these prove that the GAT can learn the GT function, but it comes as close as I can imagine without formalizing the function as a RASP + Message passing

3. A nice to have to add would be a formalized notion of the type of generalization studied (e.g. in the form of excess loss compared between base distribution and test distribution, compared to the distributional shift under question, e.g. the change in shifted reward distribution as measured by W1 vs. the shift in p(x) recovery ) could help round of the already very good precision of claims in the benchmarks. some of the studies feel more like tests of Algorithmic stability  [https://proceedings.mlr.press/v70/liu17c.html](https://proceedings.mlr.press/v70/liu17c.html)  as they don’t check a generalization error and more the performance and learnability per se

**Strengths And Weaknesses:**

Strengths

+++ Very nice construction of benchmark and ablations, truly serving as an example on how to slice and ablate a model. it seems in particular the learning curve dynamics and memorization gap under ablations and permutation of setting is valuable

++ Carefully scoped insights and conclusions, based on a *lot* of variation. (Some of the most interesting observations for me were in the appendix, e.g. the bias towards overfitting the argmax) and the rank relationship across training losses

Code and explanation of the benchmark is transparent and gives the impression that it should be easy to extend  the benchmark

Weaknesses

1. the paper could be a bit more clear on some elements and take care in the details, in particular

    1. Fig 1a) “coincidentally” shows some ablation over capacity as noted in the appendix, but this is neither quantified further (is this 3x the GAT? what parameter count? was it run to saturation? why?)

    2. There is a lot of elements floating around (graphs as states, subtrees/graphs in the flow MDP the GFN operaties on, “supervised learning” vs “distillation” vs the different TB losses, learning F,log Pf, learning these under different permutations of online/offline/interpolated, with or without destructuring…) the authors already try to make this clear, but I think having a section in the appendix which uses a combinatorial coding for each setting/combination and spells out which setting exhibited what behaviour and supports which claim would be helpful (e.g., figure 5: offline-offpolicy-neighbours-regression on unpermuted R, varying PB amongst  the set… inducing different log Pf and reward distributions=> the variation across different ) Again, the authors already take care to define all of this but I think it can be improved. ideally, for each experiment and figure, it should be possible to look up the training method, what the training loss was taking to w.r.t and how exactly the curve value is computed (JS between what and what exactly, what specific p theta are we talking about) and what claim/insight is being supported.

    3. This is most pronounce for me in a) section 4.4. to confirm my understanding: you use the correctly computed pf to regress onto, but *sample* the training samples using off policy PF, leading to something like dataset skew? or is this section using subTB in an off policy way?
        and b) the logreward distributions shown, I can’t understand those plots, are they counts with an omitted e5 number? normalized frequencies? properly labeling the plots would resolve this issue. Maybe I'm misreading the plots, but the neighbors task also doesn’t look that nice log distributed to me tbh? the counting one does, but the neighors one looks loglognormal  with an additional peak?

    4. I think the paper can actually make some more confident claims around using GAT: all the tasks are of the form “does the action lead to an increase in number of features of type “does a local neighborhood exhibit a property which is a function of a center node interacting with it’s neighbors”, the feature computation at list  is *PERFECT* for GAT and wouldn’t surprise me if it could be formalized as a RASP program [https://arxiv.org/abs/2106.06981](https://arxiv.org/abs/2106.06981). That means for this benchmark, the Hypothesis class is somewhat likely to contain the ground truth function and thus *TRUE* generalization is possible (we can’t know for sure the hypothesis class contains the GT function without constructing a GAT that will compute this as it requires also distributing the chain of  actions to the final reward, but I think the idea of using the strongest possible approximator for this is exactly what one wants)

---

> ### Author Response · Authors · 2025-02-18
>
> We thank the reviewer for their valuable feedback and questions. We are pleased to see that the reviewer found our benchmarks and ablations "very nicely constructed" and found our insights and conclusions "carefully scoped" with thorough experimental variation. We now provide detailed responses and answers to the reviewer's questions.
>
> >Fig 1a) “coincidentally” shows some ablation over capacity as noted in the appendix, but this is neither quantified further (is this 3x the GAT? what parameter count? was it run to saturation? why?)
>
> We will add further details and reasoning in our revision. You're right that this is currently unclear in the main text, and mostly hidden in the appendix. For all the architectures, what we report in Tables 1 & 2 is the usual "train with early stopping" paradigm, and we do note that more parameters seem to always help. We try to be fair between architectures by using the same numbers of parameters.
>
> >I think having a section in the appendix which uses a combinatorial coding for each setting/combination and spells out which setting exhibited what behaviour and supports which claim would be helpful
>
> Thank you for your suggestion. We did find it to be a challenge to synthesize all these results in a single paper. We will experiment with clearer tagging of results.
>
> >In section 4.4. to confirm my understanding: you use the correctly computed pf to regress onto, but sample the training samples using off policy PF, leading to something like dataset skew? or is this section using subTB in an off policy way?
>
> In 4.4 we train models with a GFlowNet objective, SubTB, as described in section 3.3. To summarize, there are two experiments in this section:
> - In the first, we train _offline_ ($\tau$ does not come from $P_F$ but from a "dataset-like" distribution); this is off-policy, except for $\mathbb{P}_\mathcal{X} \propto p(x;\theta)$
> - In the second, we train off-policy ($\tau$ comes from a modified version of $P_F$), this is _online_, since we sample $\tau$ from $s_0$ rather than from $x \in \mathcal{X}$ terminal states.
>
> >For the logreward distributions shown, I can’t understand those plots, are they counts with an omitted e5 number? normalized frequencies? properly labeling the plots would resolve this issue.
>
> They are density plots (histograms) showing the normalized count of log-rewards over our state space, but maybe you are right that it is misleading to show density plots in a discrete setting. To your point, we will amend the labeling of these plots to resolve this issue.
>
> >Maybe I'm misreading the plots, but the neighbors task also doesn’t look that nice log distributed to me tbh? the counting one does, but the neighors one looks loglognormal with an additional peak?
>
> We did not aim to have strictly log-normal looking distributions, but rather just a decent spread in the log domain that is comparable across tasks.
>
> >I think the paper can actually make some more confident claims around using GAT: all the tasks are of the form “does the action lead to an increase in number of features of type “does a local neighborhood exhibit a property which is a function of a center node interacting with it’s neighbors”, the feature computation at list  is PERFECT for GAT and wouldn’t surprise me if it could be formalized as a RASP program ... I think the idea of using the strongest possible approximator for this is exactly what one wants
>
> Thank you for the insights. We are indeed not GNN experts, so we didn't want to make any bold claim with respect to GATs, but we will look more closely into what you suggest.
>
> We again thank the reviewer for their time and effort in reviewing our paper and for their insightful comments. We believe that through this rebuttal, we will improve the overall quality of our paper and hope that we have addressed all the reviewer's questions. We are happy to address and answer any remaining points the reviewer may have.

---

### Review · Reviewer_58u1 · 2025-02-04

**Summary Of Contributions:**

This paper empirically investigates GFlowNets' ability to generalize to unvisited states during training. In particular, the authors introduce three hypotheses: (1) GFlowNets generalize only under a narrow set of distributions; (2) GFlowNets generalize because PF and F are not arbitrary, and generalization is mainly affected by the complexity of the reward. Aiming to validate these hypotheses, the authors design graph generation benchmark tasks with varying difficulty levels, expressed in terms of reward functions. The authors also discuss how GFlowNets generalize differently considering three settings: (1) distilling flow functions, (2) memorization gaps in GFlowNets, and (3) offline and off-policy training regimes.

**Audience:**

Yes

**Broader Impact Concerns:**

I don't have any concerns regarding broader impact.

**Claims And Evidence:**

Yes

**Requested Changes:**

I don't have specific requested changes. That being said, I believe that addressing my concerns above (weaknesses) would help strengthen the impact of the paper.

**Strengths And Weaknesses:**

**Strengths**

- The paper studies a relevant problem (generalization of GFlowNets) -- indeed, the favorable generalization properties have been repeatedly used in the literature to explain GFlowNet's success;

- Overall, the paper is well-written and easy-to-follow;

- The introduced benchmarks could be valuable for future research on the limitations/generalization of GFlowNets.

**Weaknesses**

- The hypothesis are somewhat vague:
  - How do we quantify the "narrow" aspect in Hypothesis 1? Isn't that important to validate the hypothesis?
  - The term structure in Hypothesis 2 is not clear  (referred to as "patterns in the data"). In addition, the paper says little about which "structures" enable GFlowNets to generalize.

- I found surprising the performance gap between GAT and other GNNs. Any idea why is this? The paper says "While they are not the simplest GNNs, they may be the most general (because of the attention mechanism)" --- is GAT more expressive than GIN? In addition, I am not aware of prior work reporting such a stark difference in performance. In fact, some studies have highlighted the limitations of the original GAT model [1,2].

- My main concern is about the generality of the findings, i.e., how broadly the conclusions apply. I wonder how much we can extrapolate the conclusions to different state graphs, training objectives, and GNNs (parametrization of the policy network). Can the authors elaborate on the dependence/sensitivity of the results on these choices?


[1] Improving Attention Mechanism in Graph Neural Networks via Cardinality Preservation, IJCAI, 2020.

[2] How attentive are graph attention networks?, ICLR, 2022.

---

> ### Author Response · Authors · 2025-02-18
>
> We thank the reviewer for their helpful feedback and insightful questions. We are delighted to see that the reviewer found our introduced benchmarks valuable for future GFN research and found our paper well written and easy-to-follow. In the following, we provide responses to the main questions raised by the reviewer.
>
> >How do we quantify the "narrow" aspect in Hypothesis 1? Isn't that important to validate the hypothesis?
>
> We agree that "narrow" is an imprecise qualifier, yet it feels like the right one to use here. In Figures 5 and 6, we observe behaviors on a fairly wide variety of sampling distributions that quickly worsen as we get away from on-policy sampling.
>
> In addition, in most existing GFN literature (that we are aware of) use of off-policy tricks requires somewhat careful tuning of the tricks' parameters. This is probably unsurprising, but also unfortunately not widely reported in papers. We've had to rely on private correspondence to verify this.
>
> >The term structure in Hypothesis 2 is not clear (referred to as "patterns in the data"). In addition, the paper says little about which "structures" enable GFlowNets to generalize.
>
> In a way, this is a trillion-dollar question! We aim to use the word _structure_ as it is commonly used in the deep learning theory & generalization literature. We again agree that it is a somewhat vague term, but one way to think clearly about structure was introduced by Zhang et al.'s 2017 paper, which we heavily rely on here.
>
> Another common way to think about structure, as we use it in Hypothesis 2, is simply to think about the fact that we perform better than a table lookup. The fact that a model can do any better than a random predictor on unseen data means that it has captured _some_ statistics of the data, i.e. structure.
>
> >I found surprising the performance gap between GAT and other GNNs. Any idea why is this?
>
> While we are not GNN experts, our intuition is that for the specific tasks we consider here, which have more to do with connectivity and counting, stacked attention mechanisms may be at some slight advantage. We do hope that readers do not extrapolate from our findings that "GATs are the best", but simply that "if a model can fit the reward, it can probably fit the policy." We initially hoped to investigate this very hypothesis, but realized that this hypothesis deserved its own entire paper.
>
> > Generality of the findings [to model choice]: can the authors elaborate on the dependence/sensitivity of the results on these choices?
>
> Considering the consistency of our results when our analysis is applied to other domains (sequences & grids, cf $\S$E) as well as other training objectives (cf $\S$D.2, $\S$D.3), we believe that our results generalize.
>
> Of course, larger scale problems will offer additional challenges and idiosyncrasies, we don't want to appear like we claim otherwise.
>
> We again thank the reviewer for their valuable feedback and their time and effort in reviewing our paper. We hope that our rebuttal answers the reviewer's questions and believe that through this, we have improved the overall quality of our work. We are more than happy to clarify any salient points that may arise.

---

### Author Response · Authors · 2025-02-18
**Feedback Summary**

We thank all the reviewers for their constructive feedback, insightful questions, and time and effort in reviewing our work, which have improved the overall quality of the paper.

The central focus of this work was to test our existing knowledge and probe generally accepted hypotheses for Generative Flow Networks (GFlowNets, GFNs) and their ability to generalize. To do this, we constructed a thorough empirical investigation, while introducing a new benchmark environment, which allowed us to probe the hypothesized generalization behaviors of GFNs in a systematic and controlled manner.

In general, all reviewers found that our experiments were well designed, thoughtful, and thorough and found our novel synthetic environment to be a valuable contribution for studying and benchmarking GFNs. Reviewers also found our paper to be generally well written, with clear and concise presentation, and transparent explanations. The primary concerns of the reviewers pertained to clarifying questions, which we address within text changes and in the individual responses.

---

### Decision · Action_Editor_c34J · 2025-03-24

**Recommendation:** Accept as is

**Comment:**

All reviewers agreed that this was an interesting and thoroughly explored piece of work.  While entirely empirical, it sheds light on the behaviour of GFNs and points towards some promising avenues for future work.

A number of minor issues and clarifications were raised by the reviewers and the authors should aim to address these in the camera ready version.

**Audience:**

Yes, anyone working on GFNs would be very interested.  Further, those working on discrete generative models broadly would likely be interested.

**Claims And Evidence:**

Yes.  As indicated by the reviewers, the paper put forward a set of synthetic problems to carefully evaluate several different hypothesis on about the generalization ability of GFNs.